# Graph Neural Dynamics via Learned Energy and Tangential Flows

## Abstract

We introduce Tango – a dynamical systems inspired framework for graph representation learning that governs node feature evolution through a learned energy landscape and its associated descent dynamics. At the core of our approach is a learnable Lyapunov function over node embeddings, whose gradient defines an energy non-increasing direction that guarantees stability. To enhance flexibility while preserving the benefits of energy-based dynamics, we incorporate a novel tangential component, learned via message passing, that evolves features while maintaining the energy value. This decomposition into orthogonal flows of energy gradient descent and tangential evolution yields a flexible form of graph dynamics, and enables effective signal propagation even in flat or ill-conditioned energy regions, that often appear in graph learning. Our method is designed to help alleviate oversquashing, and is compatible with different graph neural network backbones. Empirically, Tango achieves strong performance across a diverse set of node and graph classification and regression benchmarks, demonstrating the effectiveness of jointly learned energy functions and tangential flows for graph neural networks.

## 1 Introduction

Graph Neural Networks (GNNs) have achieved remarkable success in learning representations for graph-structured data (Bronstein et al., 2021), but they face fundamental challenges when scaling depth or modeling long-range interactions, such as vanishing gradients (Arroyo et al., 2025), over-smoothing (Nt & Maehara, 2019; Cai & Wang, 2020; Rusch et al., 2023), and over-squashing (Alon & Yahav, 2021; Topping et al., 2022; Di Giovanni et al., 2023a; Gravina et al., 2023; 2025). To address these issues, recent works have drawn connections between GNNs and dynamical systems or control theory to understand and mitigate these issues (Poli et al., 2019; Chamberlain et al., 2021b; Eliasof et al., 2021; Gravina et al., 2023; Arroyo et al., 2025). For example, treating a GNN as a continuous dynamical system (or *neural ODE*) opens the door to analyzing stability through the lens of diffusion (Chamberlain et al., 2021b), energy conservation (Rusch et al., 2022), antisymmetric dynamics Gravina et al. (2023), and Hamiltonian flows (Heilig et al., 2025). In parallel, physics-informed neural architectures have shown that embedding physical priors such as energy conservation or dissipation into neural models can dramatically improve stability and interpretability (Bhattoo et al., 2022; Gao et al., 2022; Brandstetter et al., 2022). The common theme in the aforementioned works is the reliance on the existence of *some* energy functional that is minimized or preserved by the GNN parameterization, which is often relatively simple, such as the Dirichlet energy (Rusch et al., 2023). Beyond GNNs, Lyapunov functions and Lyapunov-stable neural ODEs have also been used to guarantee stability of general neural networks (Lawrence et al., 2020; Rodriguez et al., 2022), including models designed for adversarial robustness of image classifiers where the ODE is regularized so that perturbed inputs converge to the same Lyapunov-stable equilibrium point (Kang et al., 2021), which was also studied for graph adversarial robustness in Zhao et al. (2023). In contrast, in Tango, we focus on proposing a learnable energy that is utilized with a learned gradient and tangential flow steps, and we use Lyapunov theory to derive the design of a downstream graph learning framework.

At the same time, it is well-established in bioinformatics and computational chemistry that different, and more complex, energy functions are necessary to accurately model various natural processes. For instance, in protein folding, the energy landscape is often rugged and multi-funnel-shaped, reflecting the presence of multiple stable conformations and transition pathways (Wolynes, 2005). Similarly, in

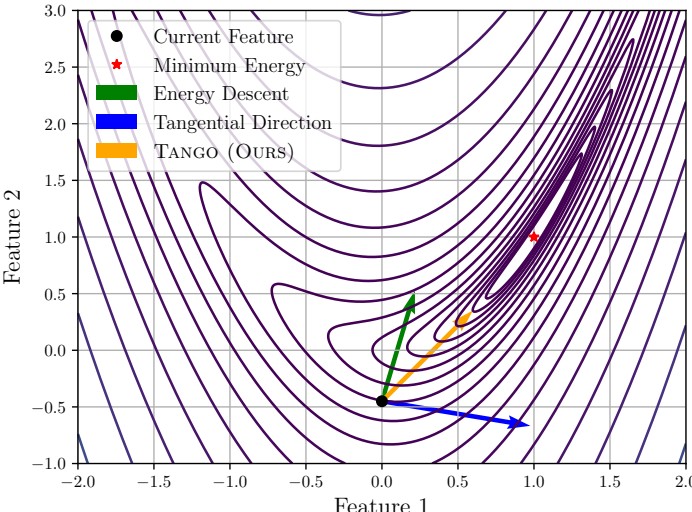

Figure 1: TANGO dynamics in a 2D feature space. We plot the level sets of a learned energy function and visualize the energy descent direction (green), the learned tangential direction (blue), and their combined vector (orange). The tangential component enables movement along level sets, while the descent component reduces energy, allowing an effective navigation of the learned energy landscape.

computational chemistry, modeling complex chemical reactions and molecular interactions requires sophisticated potential energy surfaces (Senn & Thiel, 2009).

Recently, deep learning has seen growing work on *energy-based models* (EBMs), which learn an energy function to model data distributions (e.g., images or molecules), primarily for generative modeling (LeCun et al., 2006; Xie et al., 2016; Du & Mordatch, 2019; Guo et al., 2023). In contrast, we learn a *downstream task-driven energy* whose parameters are optimized through the loss of a downstream task, such as graph or node classification, rather than generative modeling or a dedicated energy loss function.

These insights motivate a fundamental question: *How can we learn a task-driven energy function, and how can it be effectively leveraged within a GNN architecture to guide representation dynamics?* Unlike energy-based generative models, where the energy function encodes data likelihood, our focus is on learning an energy landscape whose evolution corresponds to solving a downstream task, such as node or graph classification. To address these questions, we propose to decompose feature evolution into two orthogonal components: (i) a *gradient descent* direction that minimizes the learned energy, and (ii) a *tangential direction* that evolves along its level sets, preserving energy. This structured decomposition yields a principled framework that promotes stability, enhances interpretability, and is designed to help alleviate issues such as oversquashing.

**Our Approach.** We introduce TANGO, a framework for constrained graph dynamics that incorporates a learnable Lyapunov energy function into the message-passing process, where the learned energy governs representation updates through two complementary flows: (1) an *energy descent compo-nent*, which drives convergence toward task-relevant solutions, and (2) a *tangential, conservative component*, which preserves energy while retaining flexibility by moving along energy level sets. As illustrated in Figure 1, the descent direction (green) lowers the energy, the tangential direction (blue) moves along level sets, and their combination (orange) defines the full update step, enabling effective information propagation with controlled and stable feature dynamics. TANGO's Lyapunov-inspired analysis guarantees stability in the sense of feature evolution throughout layers rather than claiming state-of-the-art performance, and our empirical studies then assess the impact of the tangential flow.

**Main Contributions.** Our contributions are as follows:

1. **Lyapunov-inspired Graph Neural Dynamics.** We introduce TANGO, a novel framework for graph representation learning that decomposes feature evolution into energy descent and tangential components, both parameterized by GNNs.

2. **Theoretical Characterization of TANGO.** We prove that, under mild assumptions, TANGO satisfies Lyapunov conditions, ensuring stable dynamics. Additionally, we show that the tangential component enables expressive yet controlled propagation and we connect this capacity to its ability to mitigate oversquashing empirically via long-range benchmarks.

3. **Strong Empirical Performance.** We evaluate TANGO on a range of graph learning benchmarks, demonstrating performance competitive with or surpassing strong and widely-used baselines.

## 2 MATHEMATICAL BACKGROUND

In this section, we provide a brief overview of Lyapunov stability theory, based on the classical treatment in Khalil & Grizzle (2002), which underpins the design of our TANGO. This theory originates from control systems and differential equations, offering a principled way to assess whether trajectories of a dynamical system remain bounded and converge over time.

**Continuous Dynamical Systems.** Let $\mathbf{h}(t) \in \mathbb{R}^d$ denote the state of a dynamical system at time $t \geq 0$, and consider a first-order ODE:

$$\frac{d\mathbf{h}(t)}{dt} = F(\mathbf{h}(t)), \tag{1}$$

where $F : \mathbb{R}^d \to \mathbb{R}^d$ is a continuous vector field. A point $\mathbf{h}^*$ is called an *equilibrium* if $F(\mathbf{h}^*) = 0$.

**Definition 1** (Lyapunov Function). *Let $\mathbf{h}^* \in \mathbb{R}^d$ be an equilibrium of the system in Equation (1). A continuously differentiable function $V : \mathbb{R}^d \to \mathbb{R}$ is called a* Lyapunov function *around $\mathbf{h}^*$ if:*

*1. $V(\mathbf{h}) \geq 0$ for all $\mathbf{h}$ in a neighborhood of $\mathbf{h}^*$, and $V(\mathbf{h}^*) = 0$;*

*2. $\frac{d}{dt}V(\mathbf{h}(t)) = \nabla_\mathbf{h} V(\mathbf{h}(t))^\top F(\mathbf{h}(t)) \leq 0$ in that neighborhood.*

The first condition ensures that $V$ is lower-bounded by 0, i.e., that value of the Lyapunov function, sometimes also referred to as *energy* is non-negative, and the second that $V$ does not increase along trajectories of the system.

We now recall a classical (Khalil & Grizzle, 2002) stability criterion for the dynamical system in Equation (1), based on the definition of a Lyapunov function, which we will later use to characterize the stability of our approach in Section 4.

**Theorem 1** (Lyapunov Stability). *Let $\mathbf{h}^*$ be an equilibrium of Equation (1) and let $V$ be a Lyapunov function in a neighborhood $\mathcal{N}$ of $\mathbf{h}^*$. If $\frac{d}{dt}V(\mathbf{h}(t)) \leq 0$ in $\mathcal{N}$, then $\mathbf{h}^*$ is* Lyapunov stable.

## 3 METHOD

As discussed in Section 1, our goal is to learn a task-driven energy function, and to devise a principled way to utilize it towards improved downstream performance for graph learning tasks, based on a combination of TANgential- and Gradient-steps Optimization of node features. We therefore call our method TANGO. In Section 3.1, we outline the blueprint of TANGO. In Section 3.2, we discuss implementation details. Later, in Section 4, we discuss the properties of our TANGO, and in Appendix C we discuss its complexity.

**Notations.** We consider a graph $\mathcal{G} = (\mathcal{V}, \mathcal{E})$ with $n = |\mathcal{V}|$ nodes and $m = |\mathcal{E}|$ edges. Let $\mathbf{H}(t) = [\mathbf{h}_1(t), \mathbf{h}_2(t), \ldots, \mathbf{h}_n(t)]^\top \in \mathbb{R}^{n \times d}$ denote the matrix of node features at continuous time $t$, where $\mathbf{h}_v(t) \in \mathbb{R}^d$ is the state of node $v$ at time $t$. Following the literature of GNNs based on dynamical systems (Eliasof et al., 2021; Gravina et al., 2023; Arroyo et al., 2025), when considering a discrete architecture with a finite number of layers, we draw an analogy between time $t$ and network depth $\ell$. Henceforth, we will interchangeably use the terms $\mathbf{H}(t)$ and $\mathbf{H}^{(\ell)}$ to denote node features at a certain time or layer of the network, depending on the context.

## 3.1 Optimizing Features with Energy Tangential and Gradient Steps

Our TANGO concept is based on a dynamical system that, given a graph energy function $V_{\mathcal{G}}$, considers two steps: (i) *energy gradient descent* and (ii) *tangential direction* flows, that evolve the node features:

$$\frac{d\mathbf{H}(t)}{dt} = \underbrace{-\alpha_{\mathcal{G}}(\mathbf{H}(t))\nabla_{\mathbf{H}}V_{\mathcal{G}}(\mathbf{H}(t))}_{\text{Energy Gradient Descent}} + \underbrace{\beta_{\mathcal{G}}(\mathbf{H}(t))\,T_{V_{\mathcal{G}}}(\mathbf{H(t)})}_{\text{Tangential Direction}}, \qquad (2)$$

where $\alpha_{\mathcal{G}}, \beta_{\mathcal{G}}$ are non-negative scalars that balance the two steps, $\nabla_{\mathbf{H}}V_{\mathcal{G}}(\mathbf{H}(t))$ is the energy gradient with respect to node features $\mathbf{H}(t)$, and $T_{V_{\mathcal{G}}}(\mathbf{H}(t))$ is an update direction that is orthogonal, i.e, tangential to the energy gradient. We note that, while in general, there are many possible directions that are orthogonal to the gradient, in Section 3.2 we specify a procedure for learning this direction. In particular, we note that, by design, the first step decreases the energy, while the second is a tangential flow that preserves energy. Below, we formalize the tangential component and provide implementation details in Section 3.2.

**Tangential Flow.** Setting $\beta_{\mathcal{G}} = 0$ in Equation (2) yields a standard energy gradient flow applied to the features. While it guarantees energy dissipation, it may suffer from slow convergence (Boyd & Vandenberghe, 2004; Nocedal & Wright, 1999) and restricted dynamics during training. As discussed in Section 1, while a gradient flow is commonly used in generative applications, accompanied by hundreds or thousands of steps are, this approach is not suitable for downstream learning, as it renders a neural network with equivalently many effective layers, that is hard to train (Peng et al., 2024) and has high computational costs. To address this, and to accelerate the minimization of the energy function, we introduce a *tangential* flow that evolves tangentially to the gradient of $V_{\mathcal{G}}$, preserving energy. As we illustrate in Figure 1, and later theoretically discuss in Section 4, while the tangential flow itself maintains the same energy level, its combination with the energy gradient descent step, as shown in Equation (2), can offer a better overall descent direction, thereby accelerating energy convergence.

In order to obtain a direction that is orthogonal to $\nabla_{\mathbf{H}}V_{\mathcal{G}}(\mathbf{H}(t))$, let $\mathbf{M}(\mathbf{H}(t))$ be a predicted update direction of the node features. We then define the *tangential* node feature update direction as:

$$T_{V_{\mathcal{G}}}(\mathbf{H}(t)) = \mathbf{M}(\mathbf{H}(t)) - \left\langle \mathbf{M}(\mathbf{H}(t)), \widehat{\nabla}_{\mathbf{H}}V_{\mathcal{G}}(\mathbf{H}(t)) \right\rangle \cdot \nabla_{\mathbf{H}}V_{\mathcal{G}}(\mathbf{H}(t)), \qquad (3)$$

where $\widehat{\nabla}_{\mathbf{H}}V_{\mathcal{G}}(\mathbf{H}(t))$ is the normalized energy gradient. Unless $\nabla_{\mathbf{H}}V_{\mathcal{G}}(\mathbf{H}(t)) = 0$, where then we define $T_{V_{\mathcal{G}}}(\mathbf{H}(t)) = \mathbf{M}(\mathbf{H}(t))$, the projection in Equation (3) removes shared the component of $\mathbf{M}(\mathbf{H}(t))$ with the energy descent direction, ensuring $T_{V_{\mathcal{G}}}$ is orthogonal to the gradient of the energy function $V_{\mathcal{G}}(\mathbf{H}(t))$.

## 3.2 Tango Graph Neural Networks

In Section 3.1, we described the concept of TANGO and its underlying continuous dynamical system. To materialize this concept and obtain a GNN, we discretize Equation (2) using the commonly used in GNNs (Gravina et al., 2023; Eliasof et al., 2021; Chamberlain et al., 2021b; Arroyo et al., 2025; Choi et al., 2023) forward Euler approach to obtain the following graph neural layer:

$$\mathbf{H}^{(\ell+1)} = \mathbf{H}^{(\ell)} + \epsilon\left(-\alpha_{\mathcal{G}}(\mathbf{H}^{(\ell)})\,\nabla_{\mathbf{H}}V_{\mathcal{G}}(\mathbf{H}^{(\ell)}) + \beta_{\mathcal{G}}(\mathbf{H}^{(\ell)})\,T_{V_{\mathcal{G}}}(\mathbf{H}^{(\ell)})\right), \qquad (4)$$

for $\ell = 0, \ldots, L-1$, where $\epsilon > 0$ is a hyperparameter step size that stems from the forward Euler discretization that is commonly used in ODE inspired GNNs (Chamberlain et al., 2021b; Eliasof et al., 2021; Gravina et al., 2023), further discussed in Appendix D.1, $\nabla_{\mathbf{H}}V_{\mathcal{G}}(\mathbf{H}^{(\ell)})$ is the gradient of the energy function defined in Equation (7). The coefficients $\alpha_{\mathcal{G}} \geq 0$, $\beta_{\mathcal{G}}$ are scalars that balance the energy descent and tangential terms, and are also predicted by the respective GNNs shown below.

**Energy Function.** We now describe the implementation of $V_{\mathcal{G}}$. Given features $\mathbf{H}^{(\ell)}$, we apply:

$$\tilde{\mathbf{H}}^{(\ell)} = \sigma\left(\text{ENERGYGNN}(\mathbf{H}^{(\ell)}; \mathcal{G})\right) \in \mathbb{R}^{n \times d}, \qquad (5)$$

where ENERGYGNN is a graph neural network (e.g., GatedGCN (Bresson & Laurent, 2018), GPS (Rampášek et al., 2022)), and $\sigma$ is a pointwise nonlinearity. We then compute per-node energy scores using a multilayer perceptron (MLP):

$$\tilde{V}_{\mathcal{G}}(\tilde{\mathbf{H}}^{(\ell)}) = \text{MLP}_{\text{E}}(\tilde{\mathbf{H}}^{(\ell)}) \in \mathbb{R}^{n \times 1}, \qquad (6)$$

and define the overall graph energy scalar value as:

$$V_{\mathcal{G}}(\mathbf{H}^{(\ell)}) = \frac{1}{n} \sum_{v \in \mathcal{V}} \tilde{V}_{\mathcal{G}}(\tilde{\mathbf{H}}^{(\ell)})_v^2 \in \mathbb{R}_{\geq 0}. \tag{7}$$

The parameters of ENERGYGNN and the energy MLP are updated through the downstream supervised loss (e.g., cross entropy or regression loss), hence $V_{\mathcal{G}}$ is a task–driven Lyapunov energy function that forms the hidden feature dynamics, where the gradient term in Equation (4) uses this learned energy function to ensure stable dynamics, while the tangential term allows feature updates along its level sets. We also note that the gradient $\nabla_{\mathbf{H}} V_{\mathcal{G}}(\mathbf{H}^{(\ell)})$ in Equation (4) is different from the gradient of the supervised loss with respect to the parameters of ENERGYGNN and TANGENTGNN. Thus, even if $\nabla_{\mathbf{H}} V_{\mathcal{G}}(\mathbf{H}^{(\ell)})$ becomes small at some layer, the parameters still receive gradients through the dependence of $V_{\mathcal{G}}$ and $T_{V_{\mathcal{G}}}$ on their weights, analogous to the outer gradient in bilevel optimization (Dempe, 2002). Empirically, we do not observe vanishing gradients when increasing depth, as shown in Appendix E.2. In addition, we employ a global sum pooling (Xu et al., 2019) to $\tilde{\mathbf{H}}^{(\ell)}$, followed by an MLP and sigmoid activation, to obtain a bounded non-negative scalar $\alpha_{\mathcal{G}}$, as follows:

$$\alpha_{\mathcal{G}}(\mathbf{H}^{(l)}) = \text{SIGMOID}\left(\text{MLP}_\alpha\left(\text{SUMPOOL}(\tilde{\mathbf{H}}^{(\ell)})\right)\right) \in [0, 1] \tag{8}$$

We note that non-negativity is required for a valid gradient descent to be obtained in Equation (4), and the bounded value is chosen to maintain stable training.

**Tangential Update.** To compute the tangential update $T_{V_{\mathcal{G}}}(\mathbf{H}^{(\ell)})$, we learn a dedicated GNN denoted by TANGENTGNN. Specifically, given input features $\mathbf{H}^{(\ell)}$, we predict a node feature update:

$$\mathbf{M}^{(\ell)} = \sigma\left(\text{TANGENTGNN}(\mathbf{H}^{(\ell)}; \mathcal{G})\right), \tag{9}$$

and define the energy-tangential component via orthogonal projection, as described in Equation (3). Also, we define the scalar $\beta_{\mathcal{G}}$ that scales the tangential term, as follows:

$$\beta_{\mathcal{G}}(\mathbf{H}^{(l)}) = \text{MLP}_\beta\left(\text{SUMPOOL}(\mathbf{M}^{(\ell)})\right) \in \mathbb{R}. \tag{10}$$

## 4 THEORETICAL PROPERTIES OF TANGO

We now analyze the continuous-time dynamics of TANGO as defined in Equation equation 2. Our analysis focuses on three aspects: *energy dissipation*, *feature evolution in flat energy landscapes*, and *the benefit of the tangent direction*. Proofs are provided in Appendix B.

**Assumptions and Notations.** Throughout this analysis, we assume that: (i) the input graph $\mathcal{G} = (\mathcal{V}, \mathcal{E})$ is connected; (ii) the energy function $V_{\mathcal{G}}(\mathbf{H}(t))$ is twice differentiable and bounded from below. For simplicity of notation, throughout this section we omit the time or layer scripts, and use the term $\mathbf{H}$ to denote node features, when possible.

We start by showing that TANGO is dissipative if $\|\nabla_{\mathbf{H}} V_{\mathcal{G}}(\mathbf{H})\|^2 > 0$, and $\alpha_{\mathcal{G}} \geq 0$ (obtained by design), corresponding to the Lyapunov stability criterion from Theorem 1.

**Proposition 1** (Energy is non-increasing). *Suppose $\alpha_{\mathcal{G}} \geq 0$ and $\|\nabla_{\mathbf{H}} V_{\mathcal{G}}(\mathbf{H})\|^2 > 0$. Then the energy $V_{\mathcal{G}}(\mathbf{H})$ is non-increasing along trajectories of Equation (2). Specifically,*

$$\begin{aligned} \frac{d}{dt} V_{\mathcal{G}}(\mathbf{H}) &= -\alpha_{\mathcal{G}}(\mathbf{H}) \|\nabla_{\mathbf{H}} V_{\mathcal{G}}(\mathbf{H})\|^2 + \beta_{\mathcal{G}}(\mathbf{H}) \langle T_{V_{\mathcal{G}}}(\mathbf{H}), \nabla_{\mathbf{H}} V_{\mathcal{G}}(\mathbf{H}) \rangle \\ &= -\alpha_{\mathcal{G}}(\mathbf{H}) \|\nabla_{\mathbf{H}} V_{\mathcal{G}}(\mathbf{H})\|^2 \leq 0. \end{aligned} \tag{11}$$

Proposition 1 establishes a standard Lyapunov property: the energy $V_{\mathcal{G}}(\mathbf{H})$ is non-increasing along trajectories (i.e., layers), and, by construction, is bounded from below. As a result, solutions remain in level sets $\{\mathbf{H} : V_{\mathcal{G}}(\mathbf{H}) \leq V_{\mathcal{G}}(\mathbf{H}^{(0)})\}$, but are not forced to collapse in feature space. In particular, Lyapunov stability obtained through the gradient flow controls the energy values, and the tangential flow component $T_{V_{\mathcal{G}}}(\mathbf{H})$ enables the evolution of node features along level sets of $V_{\mathcal{G}}$, including in regions where the energy gradient is small. We now show that unlike gradient flows, our TANGO admits evolution of node features in flat energy landscapes, a prime challenge in optimization techniques (Nocedal & Wright, 1999; Boyd & Vandenberghe, 2004).

**Proposition 2** (TANGO can Evolve Features in Flat Energy Landscapes). *Suppose $\nabla_{\mathbf{H}} V_{\mathcal{G}}(\mathbf{H}) = 0$, and $T_{V_{\mathcal{G}}}(\mathbf{H}) \neq 0$, then the* TANGO *flow in Equation (2) reads:*

$$\frac{d\mathbf{H}}{dt} = \beta_{\mathcal{G}}(\mathbf{H}) T_{V_{\mathcal{G}}}(\mathbf{H}).$$

*This implies that in* contrast *to gradient flows, the dynamics of* TANGO *obtained by the tangential term can evolve even in regions where the energy landscape is flat.*

**Theoretical Benefits of Using the Tangent Direction.** Our TANGO combines two terms as shown in Equation (2) and its discretization in Equation (4). These are the energy gradient $\nabla_{\mathbf{H}} V_{\mathcal{G}}(\mathbf{H}^{(\ell)})$ and the tangential direction vector $T_{V_{\mathcal{G}}}(\mathbf{H})$. A natural theoretical and practical question is: *under what conditions does the inclusion of the tangential direction improve over simple gradient descent?* To address this question, we first recall a classic convergence result for gradient-based minimization.

**Proposition 3** (Convergence of Gradient Descent of a Scalar Function, Nocedal & Wright (1999)). *Let $V_{\mathcal{G}}(\cdot)$ be a scalar function and let $\mathbf{H}^{(\ell+1)} = \mathbf{H}^{(\ell)} - \alpha_{\mathcal{G}}^{(\ell)}(\mathbf{H}^{(\ell)}) \nabla_{\mathbf{H}} V_{\mathcal{G}}(\mathbf{H}^{(l)})$ be a gradient-descent iteration of the energy $V_{\mathcal{G}}(\cdot)$. Then, a linear convergence is obtained, with convergence rate:*

$$r = \frac{\lambda_{\max} - \lambda_{\min}}{\lambda_{\max} + \lambda_{\min}},$$

*where $\lambda_{\max}$ is the maximal eigenvalue, and in the case of problems that involve the graph Laplacian, $\lambda_{\min}$ is the second minimal eigenvalue, i.e., the first non-zero eigenvalue of the Hessian of $V_{\mathcal{G}}(\cdot)$.*

Proposition 3 shows that gradient descent suffers in ill-conditioned problems, i.e., when the ratio between the $\lambda_{\max}$ and $\lambda_{\min}$ is large. This is common in graph-based tasks, where the Hessian may inherit poor conditioning from the graph Laplacian, particularly when oversquashing occurs due to bottlenecks in the graph Topping et al. (2022); Giraldo et al. (2023); Di Giovanni et al. (2023a). As an alternative, consider the effect of adding an orthogonal flow to the gradient descent direction. In this case, the combined update direction is

$$\mathbf{D} = \alpha_{\mathcal{G}}(\mathbf{H}^{(\ell)}) \nabla_{\mathbf{H}} V_{\mathcal{G}}(\mathbf{H}^{(\ell)}) + \beta_{\mathcal{G}}(\mathbf{H}^{(\ell)}) T_{V_{\mathcal{G}}}(\mathbf{H}^{(\ell)}). \tag{12}$$

The following proposition demonstrates that it is possible, i.e., the model has the capacity, to learn $T$ such that $\mathbf{D}$ becomes a Newton-like direction with quadratic convergence (Nocedal & Wright, 1999).

**Proposition 4** (TANGO can learn a Quadratic Convergence Direction). *Assume for simplicity that $\beta_{\mathcal{G}} = 1$, and that the Hessian of $V_{\mathcal{G}}$ is invertible. Let $\mathbf{D} = \alpha_{\mathcal{G}}(\mathbf{H}^{(\ell)}) \nabla_{\mathbf{H}} V_{\mathcal{G}}(\mathbf{H}^{(\ell)}) + T_{V_{\mathcal{G}}}(\mathbf{H}^{(\ell)})$ with $\left\langle T_{V_{\mathcal{G}}}(\mathbf{H}^{(\ell)}), \widehat{\nabla}_{\mathbf{H}} V_{\mathcal{G}}(\mathbf{H}^{(\ell)}) \right\rangle = 0$. Then, it is possible to learn a direction $T_{V_{\mathcal{G}}}(\mathbf{H}^{(\ell)})$ and a step size $\alpha_{\mathcal{G}}$ such that $\mathbf{D}$ is the Newton direction, $\mathbf{N} = (\nabla^2 V_{\mathcal{G}})^{-1} \nabla V_{\mathcal{G}}$.*

In addition to its improved global convergence, Newton's method is notable for its local convergence rate behavior, being independent of the condition number of the Hessian (Nocedal & Wright, 1999; Boyd & Vandenberghe, 2004). This implies that if the tangential flow is learned to approximate Newton direction, TANGO can overcome the slow convergence caused by highly ill-conditioned energy landscapes, as commonly observed in different second order optimization techniques and their approximations, such as conjugate gradients (CG) and LBFGS (Nocedal & Wright, 1999; Boyd & Vandenberghe, 2004). *In the context of graph learning*, Proposition 4 is particularly relevant when considering the oversquashing problem (Alon & Yahav, 2021; Di Giovanni et al., 2023a), and motivates the utilization of the conceptual blueprint of TANGO from Equation (2) for graph learning: oversquashing leads to poor conditioning; the graph Laplacian has a smallest eigenvalue of zero (for connected graphs), and the second smallest eigenvalue is also close to zero (Topping et al., 2022; Giraldo et al., 2023; Black et al., 2023; Jamadandi et al., 2024). Under these conditions, gradient flow methods, which are implicitly implemented by common GNN formulations (Di Giovanni et al., 2023b), perform poorly due to their ill-conditioned energy landscape, limiting the ability of propagating information between nodes. By enabling feature updates that can approximate second-order information, i.e., Newton-like directions, our TANGO offers a mechanism that can, in principle alleviate oversquashing. We empirically validate these results in Figure 2, where we compare our method with a Dirichlet energy gradient flow process, which is often implemented by baseline GNNs (Rusch et al., 2023; Di Giovanni et al., 2023b), with more details described in Appendix D.2, highlighting the importance of tangential flows in our TANGO, and further evaluate the effectiveness of our TANGO across oversquashing-related benchmarks in Section 5.

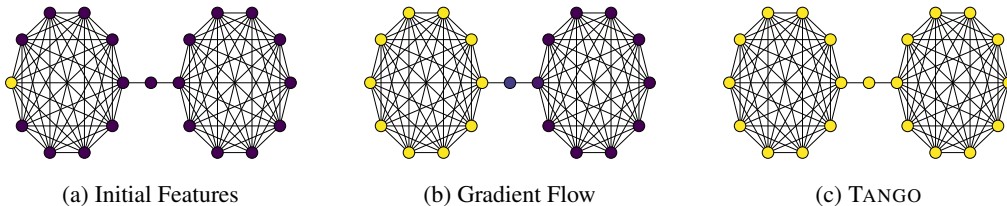

|          (a) Initial Features          |          (b) Gradient Flow          |          (c) TANGO          |

Figure 2: Comparison of propagation behaviors between gradient flow and TANGO with 50 layers. While a Dirichlet energy gradient flow struggles propagating information through the bottleneck, our TANGO is effective. We provide details on this experiment in Appendix D.2

## 5  EXPERIMENTS

We evaluate the performance of our TANGO on a suite of benchmarks: (i) synthetic benchmarks that require the exchange of messages with large distances, called graph property prediction from Gravina et al. (2023), in Section 5.1; (ii) the peptides long-range graph benchmark (Dwivedi et al., 2022b) in Section 5.3; (iii) GNN benchmarks from (Dwivedi et al., 2023) including the ZINC-12k, MNIST, CIFAR-10, PATTERN, and CLUSTER datasets; and (iv) the heterophilic node classification datasets from Platonov et al. (2023). Notably, TANGO shows consistent downstream performance improvements over its four backbone models: GCN (Kipf & Welling, 2016), GIN

Table 1: Mean test set $log_{10}(\text{MSE})(\downarrow)$ and std averaged on 4 random weight initializations on Graph Property Prediction. Lower is better. **First**, **second**, and **third** best results for each task are color-coded.

| Model | Diameter | SSSP | Eccentricity |
|---|---|---|---|
| **MPNNs** | | | |
| GatedGCN (Bresson & Laurent, 2018) | $0.1348_{\pm 0.0397}$ | $-3.2610_{\pm 0.0514}$ | $0.6995_{\pm 0.0302}$ |
| GCN (Kipf & Welling, 2016) | $0.7424_{\pm 0.0466}$ | $0.9499_{\pm 0.0001}$ | $0.8468_{\pm 0.0028}$ |
| GAT (Veličković et al., 2018) | $0.8221_{\pm 0.0752}$ | $0.6951_{\pm 0.1499}$ | $0.7909_{\pm 0.0222}$ |
| GraphSAGE (Hamilton et al., 2017) | $0.8645_{\pm 0.0401}$ | $0.2863_{\pm 0.1843}$ | $0.7863_{\pm 0.0207}$ |
| GIN (Xu et al., 2019) | $0.6131_{\pm 0.0990}$ | $-0.5408_{\pm 0.4193}$ | $0.9504_{\pm 0.0007}$ |
| GCNII (Chen et al., 2020) | $0.5287_{\pm 0.0570}$ | $-1.1329_{\pm 0.0135}$ | $0.7640_{\pm 0.0355}$ |
| **DE-GNNs** | | | |
| DGC (Poli et al., 2019) | $0.6028_{\pm 0.0050}$ | $-0.1483_{\pm 0.0231}$ | $0.8261_{\pm 0.0032}$ |
| GRAND (Chamberlain et al., 2021b) | $0.6715_{\pm 0.0490}$ | $-0.0942_{\pm 0.3897}$ | $0.6602_{\pm 0.1393}$ |
| GraphCON (Rusch et al., 2022) | $0.0964_{\pm 0.0620}$ | $-1.3836_{\pm 0.0092}$ | $0.6833_{\pm 0.0074}$ |
| A-DGN (Gravina et al., 2023) | $-0.5188_{\pm 0.1812}$ | $-3.2417_{\pm 0.0751}$ | $0.4296_{\pm 0.1003}$ |
| SWAN (Gravina et al., 2025) | $\mathbf{-0.5981}_{\pm 0.1145}$ | $-3.5425_{\pm 0.0830}$ | $\mathbf{-0.0739}_{\pm 0.2190}$ |
| PH-DGN (Heilig et al., 2025) | $-0.5385_{\pm 0.0187}$ | $-4.2993_{\pm 0.0721}$ | $-0.9348_{\pm 0.2097}$ |
| **Transformers** | | | |
| GPS (Rampášek et al., 2022) | $-0.5121_{\pm 0.0426}$ | $\mathbf{-3.5990}_{\pm 0.1949}$ | $0.6077_{\pm 0.0282}$ |
| **Ours** | | | |
| TANGO_GCN | $0.1729_{\pm 0.0382}$ | $-1.0024_{\pm 0.0854}$ | $-1.6264_{\pm 0.0053}$ |
| TANGO_GIN | $0.0433_{\pm 0.0211}$ | $-2.8923_{\pm 0.0937}$ | $-1.7228_{\pm 0.0046}$ |
| TANGO_GATEDGCN | $-0.6681_{\pm 0.0745}$ | $-5.0626_{\pm 0.0742}$ | $-1.7419_{\pm 0.0106}$ |
| TANGO_GPS | $\mathbf{-0.9772}_{\pm 0.0518}$ | $\mathbf{-5.5263}_{\pm 0.0838}$ | $\mathbf{-2.1455}_{\pm 0.0033}$ |

(Xu et al., 2019), GatedGCN (Bresson & Laurent, 2018), and GPS (Rampášek et al., 2022), highlighting its usefulness. It also offers competitive performance compared with other popular and state-of-the-art methods, such as MPNN-based models, DE-GNNs, higher-order DGNs, and graph transformers. In all experiments, TANGO is trained with the same loss function as other GNN baselines, like the cross-entropy loss. In Appendix D we provide full experimental details on the hyperparameters, benchmark evaluation, and runtimes. Additional results and comparisons, as well as evaluation on heterophilic node classification and ablation studies that isolate the effect of energy term, the tangential projection, and depth, are provided in Appendix E.

### 5.1  GRAPH PROPERTY PREDICTION

**Setup.** We consider the three graph property prediction tasks from Gravina et al. (2023), evaluating the performance of TANGO in predicting graph diameters, single source shortest paths (SSSP), and node eccentricity on synthetic graphs. To effectively address these tasks, it is essential to propagate information not only from direct neighbors but also from distant nodes within the graph. As a result, strong performance in these tasks mirrors the ability to facilitate long-range interactions.

**Results.** Table 1 reports the mean test $log_{10}(\text{MSE})$, comparing our TANGO with various MPNNs, DE-GNNs, and transformer-based models. The results highlight that TANGO, in all variants, consistently achieves the lowest (best) error across all tasks, demonstrating its efficacy compared with existing methods. For example, in the Eccentricity task, TANGO_GPS reduces the error score by over 1.2

Table 2: Test performance in five benchmarks from Dwivedi et al. (2023). Shown is the mean $_{\pm\text{std}}$ of 4 runs with different random seeds. Highlighted are the top **first**, **second**, and **third** results.

| Model | ZINC-12k | MNIST | CIFAR10 | PATTERN | CLUSTER |
|---|---|---|---|---|---|
| | MAE↓ | Accuracy↑ | Accuracy↑ | Accuracy↑ | Accuracy↑ |
| GCN (Kipf & Welling, 2016) | $0.367_{\pm0.011}$ | $90.705_{\pm0.218}$ | $55.710_{\pm0.381}$ | $71.892_{\pm0.334}$ | $68.498_{\pm0.976}$ |
| GIN (Xu et al., 2019) | $0.526_{\pm0.051}$ | $96.485_{\pm0.252}$ | $55.255_{\pm1.527}$ | $85.387_{\pm0.136}$ | $64.716_{\pm1.553}$ |
| GAT (Veličković et al., 2018) | $0.384_{\pm0.007}$ | $95.535_{\pm0.205}$ | $64.223_{\pm0.455}$ | $78.271_{\pm0.186}$ | $70.587_{\pm0.447}$ |
| GatedGCN (Bresson & Laurent, 2018) | $0.282_{\pm0.015}$ | $97.340_{\pm0.143}$ | $67.312_{\pm0.311}$ | $85.568_{\pm0.088}$ | $73.840_{\pm0.326}$ |
| PNA (Corso et al., 2020) | $0.188_{\pm0.004}$ | $97.940_{\pm0.120}$ | $70.350_{\pm0.630}$ | — | — |
| DGN (Beaini et al., 2021) | $0.168_{\pm0.003}$ | — | $\mathbf{72.838}_{\pm0.417}$ | $86.680_{\pm0.034}$ | — |
| CRaW1 (Tönshoff et al., 2023b) | $0.085_{\pm0.004}$ | $97.944_{\pm0.050}$ | $69.013_{\pm0.259}$ | — | — |
| GIN-AK+ (Zhao et al., 2022) | $0.080_{\pm0.001}$ | — | $72.190_{\pm0.130}$ | $\mathbf{86.850}_{\pm0.057}$ | — |
| SAN (Kreuzer et al., 2021a) | $0.139_{\pm0.006}$ | — | — | $86.581_{\pm0.037}$ | $76.691_{\pm0.65}$ |
| EGT (Hussain et al., 2022) | $0.108_{\pm0.009}$ | $\mathbf{98.173}_{\pm0.087}$ | $68.702_{\pm0.409}$ | $86.821_{\pm0.020}$ | $\mathbf{79.232}_{\pm0.348}$ |
| Graphormer-GD (Zhang et al., 2023) | $0.081_{\pm0.009}$ | — | — | — | — |
| GPS (Rampášek et al., 2022) | $\mathbf{0.070}_{\pm0.004}$ | $98.051_{\pm0.126}$ | $72.298_{\pm0.356}$ | $86.685_{\pm0.059}$ | $78.016_{\pm0.180}$ |
| GRIT (Ma et al., 2023) | $0.059_{\pm0.002}$ | $98.108_{\pm0.111}$ | $76.468_{\pm0.881}$ | $87.196_{\pm0.076}$ | $80.026_{\pm0.277}$ |
| TANGO$_{\text{GCN}}$ | $0.153_{\pm0.010}$ | $94.579_{\pm0.211}$ | $64.920_{\pm0.402}$ | $81.198_{\pm0.299}$ | $74.040_{\pm1.109}$ |
| TANGO$_{\text{GIN}}$ | $0.122_{\pm0.031}$ | $97.651_{\pm0.247}$ | $66.350_{\pm0.967}$ | $86.703_{\pm0.194}$ | $71.360_{\pm1.169}$ |
| TANGO$_{\text{GatedGCN}}$ | $0.128_{\pm0.011}$ | $97.788_{\pm0.105}$ | $70.894_{\pm0.329}$ | $86.672_{\pm0.071}$ | $78.194_{\pm0.307}$ |
| TANGO$_{\text{GPS}}$ | $\mathbf{0.062}_{\pm0.005}$ | $\mathbf{98.197}_{\pm0.110}$ | $75.783_{\pm0.261}$ | $87.182_{\pm0.063}$ | $80.113_{\pm0.138}$ |

points compared to PH-DGN (Heilig et al., 2025) and by over 2.0 points compared to SWAN, which are models designed to propagate information over long radii effectively. On Diameter and SSSP, TANGO$_{\text{GPS}}$ also yields gains over the strong prior DE-GNN baseline PH-DGN, improving the $\log_{10}(\text{MSE})$ by 0.4 and 1.2 points respectively. Overall, these results validate the effectiveness of our TANGO in modeling long-range interactions and and are consistent with alleviation of oversquashing. Furthermore, TANGO strengthens the performance of simple MPNN backbones like GatedGCN. For example, GatedGCN augmented with our TANGO consistently delivers better results than the baseline GatedGCN, highlighting its ability to enhance traditional MPNNs. This demonstrates that our method can effectively leverage the strengths of simple models while overcoming their limitations in long-range propagation.

## 5.2 GNN Benchmarking

**Setup.** To further evaluate the performance of our TANGO, we consider multiple GNN from Dwivedi et al. (2023), that include the *ZINC-12k* dataset, MNIST and CIFAR-10 superpixels datasets, and CLUSTER and PATTERN datasets. These datasets are commonly used to evaluate state-of-the-art techniques (Ma et al., 2023). For a fair and direct comparison with other methods, we follow the training and evaluation protocols from Dwivedi et al. (2023).

**Results.** Table 2 reports the average and standard deviation of the obtained test metric. Besides ZINC-12k, which is a regression problem with mean absolute error (MAE) as the metric, all other datasets consider the accuracy(%) metric. Our results show that across all benchmarks, our TANGO consistently improves its backbone performance, and often outperforms other strong baselines.

## 5.3 Long-Range Benchmark

**Setup.** We evaluate our method on the real-world Long-Range Graph Benchmark (LRGB) (Dwivedi et al., 2022b), focusing on *Peptides-func* and *Peptides-struct*. We follow the experimental setting in Dwivedi et al. (2022b), including the 500K parameter budget. Transformer baselines use positional and structural encodings; TANGO uses none. The datasets contain large peptide molecular graphs, whose structure and function depend on long-range interactions. Thus, short-range interactions, such as local message passing in GNNs, may be insufficient for this task.

**Results.** Table 3 provides a comparison of our TANGO model with a wide range of baselines. For example, on Peptides-struct, all TANGO variants achieve competitive MAE compared with all

methods in Table 3 under the shared parameter budget. A broader comparison is presented in Table 12. The results indicate that TANGO outperforms standard MPNNs, transformer-based GNNs, DE-GNNs, Multi-hop GNNs, and methods that use rewiring like GRAND (Chamberlain et al., 2021b) and DRew (Gutteridge et al., 2023).

## 5.4 HETEROPHILIC NODE CLASSIFICATION

**Setup.** We consider heterophilic node classification datasets; *Roman-empire, Amazon-ratings, Minesweeper, Tolokers, and Questions* tasks, to evaluate TANGO in capturing complex node relationships beyond simple homophily. We follow the training and evaluation protocols from Platonov et al. (2023).

**Results.** We report the performance of TANGO in Appendix E.1, and compare it with several recent leading methods. Specifically, we include baseline results from Finkelshtein et al. (2024); Platonov et al. (2023); Müller et al. (2024). Across all datasets, TANGO achieves competitive performance that often outperforms state-of-the-art methods, and consistently improves its backbone GNN performance, demonstrating that our TANGO can also be utilized on larger graphs and in heterophilic scenarios.

Table 3: Results for Peptides-func and Peptides-struct (3 training seeds). The first, second, and third best scores are colored.

| Model | Peptides-func AP ↑ | Peptides-struct MAE ↓ |
|---|---|---|
| **MPNNs** | | |
| GCN (Kipf & Welling, 2016) | $59.30_{\pm 0.23}$ | $0.3496_{\pm 0.0013}$ |
| GINE (Dwivedi et al., 2023) | $54.98_{\pm 0.79}$ | $0.3547_{\pm 0.0045}$ |
| GCNII (Chen et al., 2020) | $55.43_{\pm 0.78}$ | $0.3471_{\pm 0.0010}$ |
| GatedGCN (Bresson & Laurent, 2018) | $58.64_{\pm 0.77}$ | $0.3420_{\pm 0.0013}$ |
| **Multi-hop GNNs** | | |
| DIGL+MPNN+LapPE (Gasteiger et al., 2019) | $68.30_{\pm 0.26}$ | $0.2616_{\pm 0.0018}$ |
| MixHop-GCN+LapPE (Abu-El-Haija et al., 2019) | $68.43_{\pm 0.49}$ | $0.2614_{\pm 0.0023}$ |
| DRew-GCN+LapPE (Gutteridge et al., 2023) | $71.50_{\pm 0.44}$ | $0.2536_{\pm 0.0015}$ |
| **Transformers** | | |
| Transformer+LapPE (Dwivedi et al., 2023) | $63.26_{\pm 1.26}$ | $0.2529_{\pm 0.0016}$ |
| SAN+LapPE (Kreuzer et al., 2021a) | $63.84_{\pm 1.21}$ | $0.2683_{\pm 0.0043}$ |
| GPS+LapPE (Rampášek et al., 2022) | $65.35_{\pm 0.41}$ | $0.2500_{\pm 0.0005}$ |
| **DE-GNNs** | | |
| GRAND (Chamberlain et al., 2021b) | $57.89_{\pm 0.62}$ | $0.3418_{\pm 0.0015}$ |
| GraphCON (Rusch et al., 2022) | $60.22_{\pm 0.68}$ | $0.2778_{\pm 0.0018}$ |
| A-DGN (Gravina et al., 2023) | $59.75_{\pm 0.44}$ | $0.2874_{\pm 0.0021}$ |
| SWAN (Gravina et al., 2025) | $67.51_{\pm 0.39}$ | $0.2485_{\pm 0.0009}$ |
| PH-DGN (Heilig et al., 2025) | $70.12_{\pm 0.45}$ | $0.2465_{\pm 0.0020}$ |
| **Ours** | | |
| TANGO$_{\text{GCN}}$ | $69.17_{\pm 0.31}$ | $0.2432_{\pm 0.0011}$ |
| TANGO$_{\text{GIN}}$ | $68.78_{\pm 0.66}$ | $0.2440_{\pm 0.0024}$ |
| TANGO$_{\text{GATEDGCN}}$ | $68.92_{\pm 0.40}$ | $0.2451_{\pm 0.0006}$ |
| TANGO$_{\text{GPS}}$ | $70.21_{\pm 0.43}$ | $0.2422_{\pm 0.0014}$ |

## 6 RELATED WORK

We now cover two main topics related to our TANGO, with additional related works in Appendix A.

**Deep GNNs and Dynamical Systems.** A growing body of work interprets GNN layers as iterative updates in a dynamical system, providing a principled framework to analyze stability, control diffusion, and inform architectural design. Poli et al. (2019) introduced Graph Neural ODEs, inspired by neural ODEs (Ruthotto & Haber, 2020; Chen et al., 2018), modeling node feature evolution via continuous-depth ODEs aligned with graph structure, enabling adaptive computation and improved performance in dynamic settings. Similarly, Xhonneux et al. (2020) proposed Continuous GNNs, where feature channels evolve by differential equations, mitigating over-smoothing via infinite-depth limits. Follow-up works such as GODE (Zhuang et al., 2020), GRAND (Chamberlain et al., 2021b), PDE-GCN$_{\text{D}}$ (Eliasof et al., 2021), and DGC (Wang et al., 2021) view GNN layers as discrete integration steps of the heat equation to control oversmoothing (Nt & Maehara, 2019; Oono & Suzuki, 2020; Cai & Wang, 2020). Extensions like PDE-GCN$_{\text{M}}$ (Eliasof et al., 2021) and GraphCON (Rusch et al., 2022) add oscillatory components to preserve feature energy, while others leverage heat-kernel attention (Choromanski et al., 2022), anti-symmetry (Gravina et al., 2023; 2025), reaction-diffusion (Wang et al., 2023; Choi et al., 2023), advection-reaction-diffusion (Eliasof et al., 2024a) to enhance long-range or directional flow, and higher-order graph neuro ODE models (Eliasof et al., 2024b). A comprehensive overview is given in Han et al. (2023). Closely related, Di Giovanni et al. (2023b) interpret GNN layer updates as gradient flows of the Dirichlet energy, aligning message passing with energy minimization, and Zhao et al. (2023) studies the adversarial robustness of Lyapunov-based GNNs. In contrast, our TANGO takes a different approach. Instead of viewing GNN layers as discretizations whose weights are to be learned, TANGO learns a graph-adaptive,

task-driven energy and introduces a novel descent mechanism combining energy gradients with a learnable tangential component, enabling more flexible dynamics than pure gradient flows.

**Learning Energy Functions in Neural Networks.** Energy-based models (EBMs) provide a flexible framework in deep learning by learning an energy function whose low-energy regions correspond to areas with high probability for the data. They have been widely used in generative tasks such as image synthesis (LeCun et al., 2006; Xie et al., 2016; Du & Mordatch, 2019; Guo et al., 2023) and graph generation (Liu et al., 2021; Reiser et al., 2022). In contrast to these typically unsupervised settings, our work uses a *task-driven* energy function whose parameters are optimized only through supervised losses on node or graph labels, rather than via a separate generative objective. The energy plays the role of a Lyapunov potential that shapes the hidden feature dynamics; we do not require or assume that its global minima coincide with globally optimal predictions. Relatedly, Lyapunov functions, classical tools from control theory (Khalil, 2002), have been used in neural networks to ensure stable learning or inference dynamics, e.g., by enforcing stability in Neural ODEs (Rodriguez et al., 2022) and GNN-based controllers (Fallin et al., 2025), Hamiltonian graph flows for adversarial robustness (Zhao et al., 2023), as well as Beltrami flow and neural diffusion on graphs (Chamberlain et al., 2021a), which use a discretization of the Beltrami flow in joint feature and position space and induce an implicit rewiring mechanism. Notably, in a Hamiltonian system, the energy is conserved, which has been shown to be useful for adversarial robustness (Zhao et al., 2023), while a Lyapunov-stable system implies that close initial conditions evolve along similar trajectories. Lyapunov-stable neural ODEs have also been studied by regularizing an ODE to obtain Lyapunov-stable equilibria, an approach that has been found beneficial for adversarial robustness in image classification (Kang et al., 2021). Our method, TANGO, is complementary to these lines of work: it operates directly on graph-structured hidden states, learns a task-driven graph energy that is used explicitly as a Lyapunov function, and couples its gradient flow with a learned tangential component. This tangential flow can both accelerate energy minimization and maintain informative feature updates even in areas where the energy landscape is flat. In this way, TANGO bridges and extends these perspectives by introducing a graph-adaptive, task-specific energy and a novel feature evolution mechanism, which is reflected in enhanced downstream performance on graph learning tasks, as shown in Section 5.

## 7 CONCLUSIONS

We introduced TANGO, a novel framework for learning graph neural dynamics through the joint modeling of an energy descent direction and a tangential flow. By interpreting GNN message passing through the lens of Lyapunov theory and continuous dynamical systems, TANGO unifies task-driven energy-based modeling with flexible, learnable tangential flows, which allow for better utilization of the learned energy function by accelerating its minimization. We further show that the tangential component enables continued feature evolution in flat or ill-conditioned energy landscapes, offering a compelling advantage over traditional gradient flow approaches. We relate this property to the mitigation of oversquashing, a persistent challenge in graph learning. Empirically, TANGO achieves strong performance across 15 synthetic and real-world benchmarks, outperforming message-passing, diffusion-based, and attention-based GNNs. This work opens several interesting directions for future research, including the incorporation of higher-order differential operators into the tangential flow mechanism, and an analysis and regularization techniques for the learned energy landscape, as well as studying tangential flows in other domains and applications.

**Reproducibility Statement.** We will release the full codebase upon acceptance, including model implementations for TANGO backbones, training and evaluation scripts, and dataset configuration files. Comprehensive experimental details—covering dataset descriptions, splits, preprocessing, implementation specifics, parameter budgets, and runtime measurements—are provided in Appendix D.

**Ethics Statement.** This work is methodological and evaluated on public benchmark datasets that are widely used in graph learning research. We followed the licenses and terms of use for each dataset and did not collect any new human subject data. While our contribution is foundational, graph representation learning can be applied to sensitive domains. We encourage the responsible use of graph models, particularly when working with personal, social, or otherwise sensitive data. Practitioners should ensure appropriate consent and safeguards, and follow established fairness, accountability, and transparency practices.

**Usage of Large Language Models.** Large language models were used only for limited text editing suggestions. All research ideas, theoretical analysis, algorithm design, code development, experiments, and original technical writing were conducted by the authors.

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

## A  ADDITIONAL RELATED WORK

**Oversquashing in Graph Learning.** Graph neural networks (GNNs) typically operate through message-passing mechanisms, aggregating information from local neighborhoods. While effective in capturing short-range dependencies, this design often leads to *oversquashing*, a phenomenon where signals from distant nodes are compressed into fixed-size representations, impeding the flow of long-range information (Alon & Yahav, 2021; Di Giovanni et al., 2023a; Topping et al., 2022). This limitation poses a challenge in domains that demand rich global context, such as bioinformatics (Baek et al., 2021; Dwivedi et al., 2022b) and heterophilic graphs (Luan et al., 2024; Wang et al., 2024b). A range of strategies have been proposed to mitigate oversquashing. *Graph rewiring* approaches, such as SDRF (Topping et al., 2022), densify the graph to enhance connectivity prior to training. In contrast, methods like GRAND (Chamberlain et al., 2021b), BLEND (Chamberlain et al., 2021a), and DRew (Gutteridge et al., 2023) adjust the graph structure dynamically based on node features. *Transformer-based models* offer another promising route by leveraging global attention to enable direct, long-range message passing. Examples include SAN (Kreuzer et al., 2021c), Graphormer (Ying & Leskovec, 2021), and GPS (Rampášek et al., 2022), which incorporate positional encodings, such as Laplacian eigenvectors (Dwivedi et al., 2023) and random walk structural embeddings (Dwivedi et al., 2022a) to preserve structural identity. However, the quadratic complexity of full attention in these models raises scalability concerns, motivating interest in sparse attention mechanisms (Zaheer et al., 2020; Choromanski et al., 2020; Shirzad et al., 2023). An alternative line of work explores *non-local dynamics* to enhance expressivity without relying solely on attention. FLODE (Maskey et al., 2023) employs fractional graph operators, QDC (Markovich, 2023) uses quantum diffusion processes, and G2TN (Toth et al., 2022) models explicit diffusion paths to propagate information more effectively. While these approaches address the oversquashing bottleneck, they often come with increased computational demands due to dense propagation operators. For a broader overview of these techniques, see Shi et al. (2023). We note that the challenge of modeling long-range dependencies also arises in other domains, such as sequential architectures (Hochreiter & Schmidhuber, 1997; Gu et al., 2022).

**Optimization Techniques.**  The formulation of TANGO draws parallel with concepts that have been explored in the optimization literature, particularly in the design of dynamical systems that balance expressivity and convergence. While traditional gradient descent provides a robust and interpretable mechanism for minimizing energy functions, its convergence rate can be limited in poorly conditioned settings (Boyd & Vandenberghe, 2004; Nocedal & Wright, 1999), which frequently arise in graph-based problems due to structural bottlenecks (Alon & Yahav, 2021; Topping et al., 2022). Second-order approaches, such as Newton's method, are known to accelerate convergence by incorporating curvature information, albeit at increased computational cost. The combination of energy gradient descent and a learned tangential component in TANGO suggests a learnable departure from purely first-order schemes. Rather than explicitly computing or approximating the Hessian, our framework enables the model to learn corrective update directions that are orthogonal to the descent path. This design implicitly aligns with the motivations behind quasi-Newton techniques like conjugate gradients and LBFGS (Nocedal & Wright, 1999), which aim to improve convergence by leveraging directional information that complements the gradient. From this perspective, TANGO can be viewed as embedding optimization-inspired dynamics within graph learning frameworks. This is particularly relevant in scenarios affected by oversquashing (Di Giovanni et al., 2023a), where effective feature transmission often requires departing from strictly local, gradient-driven updates. By allowing energy-preserving tangential flows, TANGO introduces flexibility reminiscent of structured optimization methods, adapted to the graph learning domain.

## B  PROOFS OF THEORETICAL RESULTS

In this section, we restate the theoretical results from Section 4 and provide their proofs. As in the main text, we assume the following throughout: (i) the input graph $\mathcal{G} = (\mathcal{V}, \mathcal{E})$ is connected; (ii) the energy function $V_{\mathcal{G}}(\mathbf{H}(t))$ is twice differentiable and bounded from below. For simplicity of notation, throughout this section, we omit the time or layer scripts and use the term $\mathbf{H}$ to denote node features when possible.

**Proposition 1** (Energy Dissipation). *Suppose $\alpha_{\mathcal{G}} \geq 0$ and $\|\nabla_{\mathbf{H}} V_{\mathcal{G}}(\mathbf{H})\|^2 > 0$. Then the energy $V_{\mathcal{G}}(\mathbf{H})$ is non-increasing along trajectories of Equation equation 2. Specifically,*

$$
\begin{aligned}
\frac{d}{dt} V_{\mathcal{G}}(\mathbf{H}) &= -\alpha_{\mathcal{G}}(\mathbf{H}) \|\nabla_{\mathbf{H}} V_{\mathcal{G}}(\mathbf{H})\|^2 + \beta_{\mathcal{G}}(\mathbf{H}) \langle T_{V_{\mathcal{G}}}(\mathbf{H}), \nabla_{\mathbf{H}} V_{\mathcal{G}}(\mathbf{H}) \rangle \\
&= -\alpha_{\mathcal{G}}(\mathbf{H}) \|\nabla_{\mathbf{H}} V_{\mathcal{G}}(\mathbf{H})\|^2 \leq 0.
\end{aligned}
$$

*Proof.* By the chain rule,

$$
\frac{d}{dt} V_{\mathcal{G}}(\mathbf{H}) = \left\langle \nabla_{\mathbf{H}} V_{\mathcal{G}}(\mathbf{H}), \frac{d\mathbf{H}}{dt} \right\rangle.
$$

Substituting the dynamics of Equation equation 2:

$$
\begin{aligned}
\frac{d}{dt} V_{\mathcal{G}}(\mathbf{H}) &= \langle \nabla_{\mathbf{H}} V_{\mathcal{G}}(\mathbf{H}), -\alpha_{\mathcal{G}}(\mathbf{H}) \nabla_{\mathbf{H}} V_{\mathcal{G}}(\mathbf{H}) + \beta_{\mathcal{G}}(\mathbf{H}) T_{V_{\mathcal{G}}}(\mathbf{H}) \rangle \\
&= -\alpha_{\mathcal{G}}(\mathbf{H}) \|\nabla_{\mathbf{H}} V_{\mathcal{G}}(\mathbf{H})\|^2 + \beta_{\mathcal{G}}(\mathbf{H}) \langle T_{V_{\mathcal{G}}}(\mathbf{H}), \nabla_{\mathbf{H}} V_{\mathcal{G}}(\mathbf{H}) \rangle.
\end{aligned}
$$

As discussed in Section 3, we have by design, that

$$
\langle T_{V_{\mathcal{G}}}(\mathbf{H}), \nabla_{\mathbf{H}} V_{\mathcal{G}}(\mathbf{H}) \rangle = 0.
$$

Therefore,

$$
\frac{d}{dt} V_{\mathcal{G}}(\mathbf{H}) = -\alpha_{\mathcal{G}}(\mathbf{H}) \|\nabla_{\mathbf{H}} V_{\mathcal{G}}(\mathbf{H})\|^2.
$$

Because $\alpha_{\mathcal{G}}(\mathbf{H}) \geq 0$ by design, the energy is non-increasing, and assuming $\alpha_{\mathcal{G}}(\mathbf{H}) > 0$, the system is dissipative, i.e., its energy is decreasing. $\qquad\square$

**Proposition 2** (TANGO can Evolve Features in Flat Energy Landscapes). *Suppose $\nabla_{\mathbf{H}} V_{\mathcal{G}}(\mathbf{H}) = 0$, and $T_{V_{\mathcal{G}}}(\mathbf{H}) \neq 0$, then the* TANGO *flow in Equation (2) reads:*

$$
\frac{d\mathbf{H}}{dt} = \beta_{\mathcal{G}}(\mathbf{H}) T_{V_{\mathcal{G}}}(\mathbf{H}).
$$

*This implies that in* contrast *to gradient flows, the dynamics of* TANGO *can evolve even in regions where the energy landscape is flat.*

*Proof.* Because $\nabla_{\mathbf{H}} V_{\mathcal{G}}(\mathbf{H}) = 0$, the first term in Equation (2) vanishes, and the TANGO dynamical system reads:

$$
\frac{d\mathbf{H}}{dt} = \beta_{\mathcal{G}}(\mathbf{H}) T_{V_{\mathcal{G}}}(\mathbf{H}),
$$

Assuming that $T_{V_{\mathcal{G}}}(\mathbf{H}) \neq 0$, TANGO can continue evolving node features also in cases where $\nabla_{\mathbf{H}} V_{\mathcal{G}}(\mathbf{H}) = 0$, i.e., where the energy landscape is flat. $\qquad\square$

**Proposition 3** (Convergence of Gradient Descent of a Scalar Function, Nocedal & Wright (1999)). *Let $V_{\mathcal{G}}(\cdot)$ be a scalar function and let $\mathbf{H}^{(\ell+1)} = \mathbf{H}^{(\ell)} - \alpha_{\mathcal{G}}^{(\ell)}(\mathbf{H}^{(\ell)}) \nabla_{\mathbf{H}} V_{\mathcal{G}}(\mathbf{H}^{(l)})$ be a gradient-descent iteration of the energy $V_{\mathcal{G}}(\cdot)$. Then, a linear convergence is obtained, with convergence rate:*

$$
r = \frac{\lambda_{\max} - \lambda_{\min}}{\lambda_{\max} + \lambda_{\min}},
$$

*where $\lambda_{\max}$ is the maximal eigenvalue, and in the case of problems that involve the graph Laplacian, $\lambda_{\min}$ is the second minimal eigenvalue, i.e., the first non-zero eigenvalue of the Hessian of $V_{\mathcal{G}}(\cdot)$.*

**Proposition 4** (TANGO can learn a Quadratic Convergence Direction). *Assume for simplicity that $\beta_{\mathcal{G}} = 1$, and that the Hessian of $V_{\mathcal{G}}$ is invertible. Let $\mathbf{D} = \alpha_{\mathcal{G}}(\mathbf{H}^{(\ell)}) \nabla_{\mathbf{H}} V_{\mathcal{G}}(\mathbf{H}^{(\ell)}) + T_{V_{\mathcal{G}}}(\mathbf{H}^{(\ell)})$ with $\left\langle T_{V_{\mathcal{G}}}(\mathbf{H}^{(\ell)}), \widehat{\nabla}_{\mathbf{H}} V_{\mathcal{G}}(\mathbf{H}^{(\ell)}) \right\rangle = 0$. Then, it is possible to learn a direction $T_{V_{\mathcal{G}}}(\mathbf{H}^{(\ell)})$ and a step size $\alpha_{\mathcal{G}}$ such that $\mathbf{D}$ is the Newton direction, $\mathbf{N} = (\nabla^2 V_{\mathcal{G}})^{-1} \nabla V_{\mathcal{G}}$.*

*Proof.* We aim to construct a direction $\mathbf{D} = \alpha_{\mathcal{G}}(\mathbf{H}) \, \nabla_{\mathbf{H}} V_{\mathcal{G}}(\mathbf{H}) + T_{V_{\mathcal{G}}}(\mathbf{H})$ that matches the Newton direction:

$$\mathbf{N} = \left(\nabla_{\mathbf{H}}^2 V_{\mathcal{G}}(\mathbf{H})\right)^{-1} \nabla_{\mathbf{H}} V_{\mathcal{G}}(\mathbf{H}).$$

Recall that by design, we have that $T_{V_{\mathcal{G}}}(\mathbf{H})$ is orthogonal to the energy gradient, i.e., $\langle T_{V_{\mathcal{G}}}(\mathbf{H}), \, \nabla_{\mathbf{H}} V_{\mathcal{G}}(\mathbf{H}) \rangle = 0$. Then, we can express a Newton direction by the decomposition:

$$\mathbf{N} = \alpha_{\mathcal{G}}(\mathbf{H}) \, \nabla_{\mathbf{H}} V_{\mathcal{G}}(\mathbf{H}) + T_{V_{\mathcal{G}}}(\mathbf{H}).$$

Solving for the orthogonal component yields:

$$T_{V_{\mathcal{G}}}(\mathbf{H}) = \mathbf{N} - \alpha_{\mathcal{G}}(\mathbf{H}) \, \nabla_{\mathbf{H}} V_{\mathcal{G}}(\mathbf{H}).$$

To enforce orthogonality, we require:

$$\langle \mathbf{N} - \alpha_{\mathcal{G}}(\mathbf{H}) \, \nabla_{\mathbf{H}} V_{\mathcal{G}}(\mathbf{H}), \, \nabla_{\mathbf{H}} V_{\mathcal{G}}(\mathbf{H}) \rangle = 0.$$

Expanding and simplifying, we find:

$$\langle \mathbf{N}, \, \nabla_{\mathbf{H}} V_{\mathcal{G}}(\mathbf{H}) \rangle - \alpha_{\mathcal{G}}(\mathbf{H}) \, \|\nabla_{\mathbf{H}} V_{\mathcal{G}}(\mathbf{H})\|^2 = 0,$$

and the optimal step size is given by:

$$\alpha_{\mathcal{G}}(\mathbf{H}) = \frac{\langle \mathbf{N}, \, \nabla_{\mathbf{H}} V_{\mathcal{G}}(\mathbf{H}) \rangle}{\|\nabla_{\mathbf{H}} V_{\mathcal{G}}(\mathbf{H})\|^2},$$

showing that it is possible to learn a Newton direction, i.e., a quadratic energy convergence direction. $\square$

## C    COMPLEXITY AND RUNTIMES

**Complexity.** Each step of TANGO requires computing the gradient of the learned energy function $V_{\mathcal{G}}(\mathbf{H}^{(\ell)})$, that is defined in Equation (7). This involves two main operations: (i) forward and backward passes through the energy network ENERGYGNN, which contains $L_{\text{energy}}$ message-passing layers and an MLP; and (ii) automatic differentiation to compute $\nabla_{\mathbf{H}} V_{\mathcal{G}}(\mathbf{H}^{(\ell)})$ with respect to the input node features. In parallel, the tangential flow direction $T_{V_{\mathcal{G}}}(\mathbf{H}^{(\ell)})$ is obtained by projecting the vector field $\mathbf{M}^{(\ell)}$ computed by a separate TANGENTGNN with $L_{\text{tangent}}$ layers onto the orthogonal complement of the normalized energy gradient, as shown in Equation (3). This projection is of computational cost of $O(nd)$ per step, where $n = |\mathcal{V}|$ and $d$ is the feature dimensionality. In addition, scalar coefficients $\alpha_{\mathcal{G}}$ and $\beta_{\mathcal{G}}$ are computed from pooled node features using MLPs (Equations (8) and (10)). Assuming both ENERGYGNN and TANGENTGNN are message-passing architectures with linear complexity in the number of nodes and edges, and setting $L_{\text{energy}} = L_{\text{tangent}}$, the total complexity per layer becomes $O(L_{\text{gnn}} \cdot (n + m) \cdot d)$, where $L_{\text{gnn}}$ is the number of GNN layers used in each subnetwork and $m = |\mathcal{E}|$ is the number of edges. Unrolling the dynamics over $L$ steps, the overall computational complexity of TANGO is:

$$O\left(L \cdot L_{\text{gnn}} \cdot (|\mathcal{V}| + |\mathcal{E}|) \cdot d\right).$$

**Memory.** The memory footprint of TANGO is dominated by storing activations for backpropagation, as in any deep GNN. Using two subnetworks of the same backbone type (ENERGYGNN and TANGENTGNN) roughly doubles the number of feature tensors that need to be kept in memory. Nonetheless, we match the overall parameter budget to the underlying backbone by reducing widths where needed (see Table 7), so that the resulting models remain comparable in size. The asymptotic memory complexity remains linear in the number of nodes and edges and in the number of unrolled steps: the additional cost of computing $\nabla_{\mathbf{H}} V_{\mathcal{G}}(\mathbf{H}^{(\ell)})$ is handled by standard automatic differentiation, which is already widely used in graph models that differentiate through intermediate node states or energies. In practice, we did not observe memory blow-up or numerical instabilities due to the projection in Equation (3); all reported configurations train within the same hardware constraints as the corresponding backbones.

**Parameter count comparison.**    To ensure a fair comparison, we match the parameter budget of each backbone when instantiating TANGO. Table 4 reports parameter counts alongside mean performance and standard deviation across datasets and metrics. As shown, TANGO uses a comparable number of parameters to its corresponding backbones while achieving consistently stronger results. This protocol allows us to isolates the contribution of our Lyapunov-guided dynamics in TANGO from the number of parameters.

Table 4: Comparison of models across datasets. Performance is reported as mean $\pm$ standard deviation, with the metric indicated; $\downarrow$ means lower is better and $\uparrow$ means higher is better.

| Dataset | Model | Params | Performance (metric) |
|---|---|---|---|
| ZINC-12k | GatedGCN | 503,013 | $0.282 \pm 0.015$ (MAE $\downarrow$) |
| | TANGO | 503,409 | $0.128 \pm 0.011$ (MAE $\downarrow$) |
| | GPS | 423,717 | $0.070 \pm 0.004$ (MAE $\downarrow$) |
| | TANGO | 422,947 | $0.062 \pm 0.005$ (MAE $\downarrow$) |
| Roman-Empire | GatedGCN | 541,086 | $74.46 \pm 0.54$ (Acc $\uparrow$) |
| | TANGO | 520,822 | $91.89 \pm 0.30$ (Acc $\uparrow$) |
| | GPS | 524,218 | $87.04 \pm 0.58$ (Acc $\uparrow$) |
| | TANGO | 525,016 | $91.08 \pm 0.57$ (Acc $\uparrow$) |
| Peptides-func | GatedGCN | 496,184 | $58.64 \pm 0.77$ (AP $\uparrow$) |
| | TANGO | 496,590 | $68.92 \pm 0.40$ (AP $\uparrow$) |
| | GPS | 504,362 | $65.35 \pm 0.41$ (AP $\uparrow$) |
| | TANGO | 502,938 | $70.21 \pm 0.43$ (AP $\uparrow$) |

Table 5: Training runtime comparison per epoch (ms) across datasets and baselines. TANGO achieves a similar runtime to other methods. We note that while TANGO requires more computation time than its backbone GNN, it remains efficient and within the same order of magnitude of computations as other methods, while offering improved performance as shown in the results in Tables 1, 4 and 9.

| Model | Questions | Roman-Empire | ZINC-12k | Diameter |
|---|---|---|---|---|
| GIN | 108.72 | 23.32 | 382.63 | 450.21 |
| GCN | 69.77 | 14.96 | 249.45 | 294.35 |
| GatedGCN | 129.92 | 27.86 | 453.76 | 537.57 |
| GAT | 112.40 | 24.12 | 398.02 | 471.40 |
| GPS | 429.08 | 92.08 | 1506.05 | 1822.03 |
| GRIT | 520.00 | 111.57 | 1865.06 | 2163.81 |
| TANGO-GatedGCN | 184.98 | 39.66 | 653.29 | 778.22 |
| TANGO-GPS | 694.27 | 148.96 | 2435.85 | 2899.24 |

**Runtimes.** We benchmark training runtimes per iteration for TANGO instantiated on two backbones (GatedGCN and GPS) and compare against standard baselines across four datasets: Questions, Roman-Empire, ZINC-12k, and Diameter. The measurements are reported in Table 5. It is evident that TANGO introduces a moderate overhead relative to its corresponding backbone while remaining in the same order of magnitude as commonly used architectures. In particular, TANGO-GatedGCN is slower than GatedGCN but substantially faster than GPS-class methods, and TANGO-GPS scales proportionally with GPS. All measurements were taken under matched hyperparameters with 256 channels, 8 layers on a single NVIDIA RTX6000 Ada GPU with 48 GB memory.

## D EXPERIMENTAL DETAILS

In this section, we provide additional experimental details.

**Computational Resources.** Our experiments are run on NVIDIA RTX6000 Ada with 48GB of memory. Our code is implemented in PyTorch Paszke et al. (2019), and will be publicly released upon acceptance.

**Baselines.** We consider different classical and state-of-the-art GNN baselines. Specifically:

- Classical MPNNs, i.e., GCN (Kipf & Welling, 2016), GraphSAGE (Hamilton et al., 2017), GAT (Veličković et al., 2018), GatedGCN (Bresson & Laurent, 2018), GIN (Xu et al., 2019), GINE (Hu et al., 2020), GCNII (Chen et al., 2020), and CoGNN (Finkelshtein et al., 2024);

- Heterophily-specific models, i.e., H2GCN (Zhu et al., 2020), CPGNN (Zhu et al., 2021), FAGCN (Bo et al., 2021), GPR-GNN (Chien et al., 2021), FSGNN (Maurya et al., 2022), GloGNN Li et al. (2022), GBK-GNN (Du et al., 2022), and JacobiConv (Wang & Zhang, 2022);

- DE-DGNs, i.e., DGC (Wang et al., 2021), GRAND (Chamberlain et al., 2021b), Graph-CON (Rusch et al., 2022), A-DGN (Gravina et al., 2023), and SWAN (Gravina et al., 2025);

- Graph Transformers, i.e., Transformer (Vaswani et al., 2017; Dwivedi & Bresson, 2021), GT (Shi et al., 2021), SAN (Kreuzer et al., 2021b), GPS (Rampášek et al., 2022), GOAT (Kong et al., 2023), and Exphormer (Shirzad et al., 2023);

- Higher-Order DGNs, i.e., DIGL (Gasteiger et al., 2019), MixHop (Abu-El-Haija et al., 2019), and DRew (Gutteridge et al., 2023).

- SSM-based GNN, i.e., Graph-Mamba (Wang et al., 2024a), GMN (Behrouz & Hashemi, 2024), and GPS+Mamba (Behrouz & Hashemi, 2024)

## D.1 FORWARD EULER DISCRETIZATION AND STABILITY

Recall that the continuous-time dynamics of TANGO are given by Equation (2) where $V_{\mathcal{G}}$ is a non-negative Lyapunov energy, $\nabla_{\mathbf{H}} V_{\mathcal{G}}$ is its gradient with respect to node features, and $T_{V_{\mathcal{G}}}$ is constrained to be orthogonal to this gradient. In discrete depth, we implement TANGO using the forward Euler residual update as shown in Equation (4) for $\ell = 0, \ldots, L-1$, with step size $\epsilon > 0$. In classical numerical analysis, explicit Euler schemes can be fragile when applied to a *fixed* stiff ODE: for a given vector field, only a restricted range of step sizes yields stable trajectories. In TANGO, however, we do not discretize a predetermined ODE. Instead, the vector field is *learned*, while $\epsilon$ is selected from a small grid. If a particular combination of parameters and step size led to severe instability (for example exploding feature norms or chaotic trajectories), the resulting network would fail to train and would not attain the reported validation and test performance. The fact that TANGO trains reliably across all benchmarks, including deep networks (as shown in Appendix E.2), empirically indicates that the learned dynamics lie in a regime where the forward-Euler discretization is numerically well behaved.

Our continuous-time analysis in Section 4 shows that, under the assumptions of Proposition 1, the energy $V_{\mathcal{G}}(\mathbf{H}(t))$ is non-increasing along trajectories of Equation (2). The forward Euler update Equation (4) inherits this behavior up to second-order terms in $\epsilon$. Let us assume that $V_{\mathcal{G}}$ is twice continuously differentiable and that its Hessian is bounded in operator norm by $L_V$ on the subset of feature space visited during training. A second-order Taylor expansion of $V_{\mathcal{G}}$ along the direction $\epsilon F(\mathbf{H}^{(\ell)})$ gives

$$V_{\mathcal{G}}(\mathbf{H}^{(\ell+1)}) = V_{\mathcal{G}}(\mathbf{H}^{(\ell)}) + \epsilon \left\langle \nabla_{\mathbf{H}} V_{\mathcal{G}}(\mathbf{H}^{(\ell)}), F(\mathbf{H}^{(\ell)}) \right\rangle + \frac{\epsilon^2}{2} F(\mathbf{H}^{(\ell)})^\top \nabla_{\mathbf{H}}^2 V_{\mathcal{G}}(\boldsymbol{\xi}^{(\ell)}) F(\mathbf{H}^{(\ell)}),$$
(13)

for some intermediate point $\boldsymbol{\xi}^{(\ell)}$ on the line segment between $\mathbf{H}^{(\ell)}$ and $\mathbf{H}^{(\ell+1)}$. By construction, $T_{V_{\mathcal{G}}}$ is orthogonal to $\nabla_{\mathbf{H}} V_{\mathcal{G}}$, and $\alpha_{\mathcal{G}}(\mathbf{H}^{(\ell)}) \geq 0$, so the inner product term satisfies

$$\left\langle \nabla_{\mathbf{H}} V_{\mathcal{G}}(\mathbf{H}^{(\ell)}), F(\mathbf{H}^{(\ell)}) \right\rangle = -\alpha_{\mathcal{G}}(\mathbf{H}^{(\ell)}) \left\| \nabla_{\mathbf{H}} V_{\mathcal{G}}(\mathbf{H}^{(\ell)}) \right\|^2 \leq 0.$$

Using the Hessian bound and the Cauchy–Schwarz inequality, the remainder term is bounded above by

$$\frac{\epsilon^2}{2} F(\mathbf{H}^{(\ell)})^\top \nabla_{\mathbf{H}}^2 V_{\mathcal{G}}(\boldsymbol{\xi}^{(\ell)}) F(\mathbf{H}^{(\ell)}) \leq \frac{L_V}{2} \epsilon^2 \left\| F(\mathbf{H}^{(\ell)}) \right\|^2.$$

Combining these two observations, we obtain

$$V_{\mathcal{G}}(\mathbf{H}^{(\ell+1)}) \leq V_{\mathcal{G}}(\mathbf{H}^{(\ell)}) + \frac{L_V}{2} \epsilon^2 \left\| F(\mathbf{H}^{(\ell)}) \right\|^2,$$
(14)

which shows that the discrete-time dynamics are Lyapunov-dissipative up to a second-order term in the step size. In the limit $\epsilon \to 0$, the discrete trajectories converge to those of the continuous-time system and inherit its strict energy dissipation. In practice, we select $\epsilon$ from a range where $V_{\mathcal{G}}(\mathbf{H}^{(\ell)})$ is empirically non-increasing along depth and training remains stable.

This behavior is analogous to what is observed in residual networks and continuous-depth models. Architectures such as Neural Ordinary Differential Equations for Euclidean data and graph-based ODE models such as in Chamberlain et al. (2021b); Eliasof et al. (2021); Gravina et al. (2023); Rusch et al. (2022); Choi et al. (2023) are typically implemented using an explicit Euler or closely related explicit Runge–Kutta scheme on a *learned* vector field. In all these cases, the effective stability of the discretization is governed by the interaction between the learned dynamics, the chosen step size, and the training objective: unstable combinations do not converge during training, while successful training implicitly identifies a regime where the explicit integrator is adequate. Our contribution is orthogonal to the specific choice of time integrator. TANGO proposes a Lyapunov-structured vector field on node embeddings, decomposed into an energy-dissipating gradient term and an energy-preserving tangential term, and instantiates it with the same forward Euler step that is standard in ODE-inspired GNNs (Chamberlain et al., 2021b; Eliasof et al., 2021; Gravina et al., 2023). More sophisticated integrators (for example implicit–explicit schemes, semi-implicit methods, or higher-order Runge–Kutta rules) could in principle be combined with TANGO as well, and a systematic comparison of such integrators in this setting is an interesting avenue for future work.

## D.2 SYNTHETIC EXAMPLE

In the synthetic example in Figure 2, we demonstrate the effectiveness of TANGO in overcoming the oversquashing issue in GNNs. To do that, we consider a Barbell graph, where all node features are set to 0, besides the left-most node in the graph, which is set to 1, as shown in Figure 2(a). The goal is to allow the information to propagate through all nodes effectively. We do this by considering a gradient flow process of the Dirichlet energy using 50 layers (steps), as shown in Figure 2(b), where it is noticeable that the information is now flowing to the right part in the graph, because of the bottleneck between the two cliques. However, as we show in Figure 2(c), by considering our TANGO, which utilizes both an energy flow as well as a tangential flow, it is possible to effectively propagate the information through all the nodes in the graphs.

## D.3 GRAPH PROPERTY PREDICTION

**Dataset.** We construct our benchmark following the protocol introduced by Gravina et al. (2023). Graph instances are synthetically generated from a variety of canonical topologies, including Erdős–Rényi, Barabasi-Albert, caveman, tree, and grid models. Each graph consists of 25 to 35 nodes, with node features initialized as random identifiers sampled uniformly from the interval $[0, 1)$. The prediction targets encompass several structural tasks: computing the shortest paths from a source node, estimating node eccentricity, and determining graph diameter. The complete dataset contains 7,040 graphs, split into 5,120 for training, 640 for validation, and 1,280 for testing. These tasks inherently demand capturing long-range dependencies, as they involve global graph computations such as shortest path inference. As highlighted in Gravina et al. (2023), traditional algorithms like Bellman-Ford or Dijkstra's method require multiple rounds of message propagation, which motivates the need for expressive graph models. The benchmark graph families, such as caveman, tree, line, star, caterpillar, and lobster, frequently include structural bottlenecks that are known to induce over-squashing effects (Topping et al., 2022), posing additional challenges for message-passing-based GNNs.

**Experimental Setup.** We adopt the same evaluation framework as Gravina et al. (2023), including datasets, training routines, and hyperparameter spaces. Model training is conducted using the Adam optimizer for up to 1500 epochs, with early stopping triggered after 100 consecutive epochs of no improvement on the validation Mean Squared Error (MSE). Hyperparameters are selected via grid search, and performance is averaged over 4 independent runs with different random seeds for weight initialization. A summary of the hyperparameter grid used in our experiments is provided in Table 7.

## D.4 GRAPH BENCHMARKS

**Dataset.** To comprehensively assess the capabilities of TANGO, we evaluate its performance on a diverse set of graph learning benchmarks curated by Dwivedi et al. (2023). The benchmark suite includes: *ZINC-12k*, a molecular regression dataset containing chemical compounds, where the goal is to predict the constrained solubility of each molecule. Graphs represent molecular structures, with atoms as nodes and chemical bonds as edges. Node and edge features encode atom types and bond

types, respectively. *MNIST* and *CIFAR-10* superpixels are graph-structured versions of standard image classification datasets, where images are converted into sparse graphs of superpixels. Each superpixel forms a node, and edges are based on spatial adjacency. The tasks involve classifying digits (MNIST) and natural objects (CIFAR-10) based on graph-structured representations. *CLUSTER* and *PATTERN* are synthetic datasets designed to assess the relational inductive biases of graph neural networks. Both datasets are generated from a set of stochastic block models (SBMs). In *CLUSTER*, the task is to group nodes by community, while *PATTERN* involves identifying specific structural patterns within each graph. These datasets span a variety of domains: chemical, image, and synthetic graphs, and are commonly used to benchmark architectural innovations in GNNs (Ma et al., 2023). We follow the official training, validation, and test splits provided by Dwivedi et al. (2023), ensuring consistency in evaluation across models.

**Experimental Setup.** We adhere to the training and evaluation protocol established in Dwivedi et al. (2023). For each dataset, we perform hyperparameter tuning via grid search, optimizing the corresponding evaluation metrics: Mean Absolute Error (MAE) for *ZINC-12k*, and classification accuracy for the remaining tasks. We use the AdamW optimizer and train all models for up to 300 epochs, with early stopping based on validation performance. To ensure comparability with prior work, we respect the same parameter budgets used in the original benchmark and maintain the architectural constraints defined for fair evaluation. Each configuration is trained with three random seeds, and we report the average and standard deviation of the results. Hyperparameter ranges used in this set of experiments are summarized in Table 7.

### D.5 LONG RANGE GRAPH BENCHMARK

**Dataset.** To evaluate model performance on real-world graphs with significant long-range dependencies, we utilize the *Peptides-func* and *Peptides-struct* benchmarks introduced in Dwivedi et al. (2022b). These datasets represent peptide molecules as graphs, where nodes correspond to heavy (non-hydrogen) atoms, and edges denote chemical bonds. *Peptides-func* is a multi-label classification task with 10 functional categories, including antibacterial, antiviral, and signaling-related properties. In contrast, *Peptides-struct* focuses on regression, targeting physical and geometric attributes such as molecular inertia (weighted by atomic mass and valence), atom pair distance extremes, sphericity, and average deviation from a best-fit plane. Together, the two datasets comprise 15,535 peptide graphs and roughly 2.3 million nodes. We adopt the official train/validation/test partitions from Dwivedi et al. (2022b) and report mean and standard deviation across three different random seeds for each experiment.

**Experimental Setup.** We follow the evaluation protocol established in Dwivedi et al. (2022b), including dataset usage, training strategy, and model capacity constraints. Hyperparameter tuning is carried out via grid search, optimizing for Average Precision (AP) in the classification task and Mean Absolute Error (MAE) in the regression task. All models are trained using the AdamW optimizer for up to 300 epochs, with early stopping based on validation performance. To ensure fairness and comparability, all models adhere to the 500K parameter limit, in line with the settings of Dwivedi et al. (2022b) and Gutteridge et al. (2023). Each configuration is run three times with different weight initializations, and the results are averaged. Details of the hyperparameter ranges considered can be found in Table 7.

### D.6 HETEROPHILIC NODE CLASSIFICATION

**Dataset.** For evaluating performance in heterophilic graph settings, we consider five benchmark tasks introduced by Platonov et al. (2023): *Roman-Empire*, *Amazon-Ratings*, *Minesweeper*, *Tolokers*, and *Questions*. These datasets span a diverse range of domains and graph topologies. *Roman-Empire* is constructed from the Wikipedia article on the Roman Empire, where nodes represent words and edges capture either sequential adjacency or syntactic relations. The task is node classification with 18 syntactic categories, and the underlying graph is sparse and chain-structured, suggesting the presence of long-range dependencies. *Amazon-Ratings* originates from Amazon's product co-purchasing graph. Nodes correspond to products, linked if they are frequently bought together. The classification task involves predicting discretized average product ratings (five classes), with node features derived from fastText embeddings of product descriptions. *Minesweeper* is a synthetic dataset modeled as a $100 \times 100$ grid. Nodes represent individual cells, with edges connecting adjacent cells. A random

20% of nodes are labeled as mines, and the objective is to classify mine-containing cells based on one-hot features that encode the number of neighboring mines. *Tolokers* is based on the Toloka crowdsourcing platform (Likhobaba et al., 2023), where each node is a worker (toloker), and edges indicate co-participation on the same project. The task involves binary classification to detect whether a worker has been banned, using node features from user profiles and performance metrics. *Questions* draws from user interaction data on Yandex Q, a question-answering forum. Nodes represent users, and edges capture answering interactions. The goal is to identify users who remain active, with input features derived from user-provided descriptions. A summary of dataset statistics is provided in Table 6.

Table 6: Statistics of the heterophilic node classification datasets.

|  | Roman-empire | Amazon-ratings | Minesweeper | Tolokers | Questions |
|---|---|---|---|---|---|
| N. nodes | 22,662 | 24,492 | 10,000 | 11,758 | 48,921 |
| N. edges | 32,927 | 93,050 | 39,402 | 519,000 | 153,540 |
| Avg degree | 2.91 | 7.60 | 7.88 | 88.28 | 6.28 |
| Diameter | 6,824 | 46 | 99 | 11 | 16 |
| Node features | 300 | 300 | 7 | 10 | 301 |
| Classes | 18 | 5 | 2 | 2 | 2 |
| Edge homophily | 0.05 | 0.38 | 0.68 | 0.59 | 0.84 |

**Experimental Setup.** Our experimental procedure aligns with that of Freitas et al. (2021) and Platonov et al. (2023). We conduct a grid search to optimize model performance, using classification accuracy for the *Roman-Empire* and *Amazon-Ratings* tasks, and ROC-AUC for *Minesweeper*, *Tolokers*, and *Questions*. Each model is trained using the AdamW optimizer for a maximum of 300 epochs. Our experiments follow the official dataset splits provided by Platonov et al. (2023). For each model configuration, we perform multiple training runs with different random seeds and report the mean and standard deviation of the results. The hyperparameter grid explored in these experiments is summarized in Table 7.

### D.7 HYPERPARAMETERS

In Table 7, we summarize the hyperparameter grids used for tuning our TANGO across different benchmarks. In particular, we have followed similar practices from the literature (Gravina et al., 2025; Rusch et al., 2022). Alongside standard training hyperparameters such as learning rate, weight decay, and batch size, our method introduces several additional components. These include the number of unrolled steps $L$ (corresponding to the depth of the energy-based dynamics), the hidden dimension $d$ of node features, and the number of message-passing layers $L_{gnn}$ used within the internal ENERGYGNN and TANGENTGNN modules. In all experiments, we share the architecture depth between ENERGYGNN and TANGENTGNN. We also tune the step size $\epsilon$ used in the forward Euler update (Equation (4)), which controls the integration scale of the continuous dynamics. We explore multiple values of $L$ to assess how the number of dynamical steps impacts long-range propagation across different tasks. Details of the complete hyperparameter grid can be found in Table 7.

Table 7: Hyperparameter grids used during model selection for the different benchmark categories: *GraphPropPred* (Diameter, SSSP, Eccentricity), *LRGB* (Peptides-func/struct), *Graph Benchmarks* (ZINC-12k, MNIST, CIFAR-10, CLUSTER, PATTERN), and *Node Classification* (Roman-Empire, Amazon-Ratings, Minesweeper, Tolokers, Questions).

| Hyperparameter | GraphPropPred | LRGB | Graph Benchmarks | Node Classification |
|---|---|---|---|---|
| Unrolled steps $L$ | {1,5,10,20} | {2,4,8,16,32} | {2,4,8,16,32} | {2,4,8,16,32} |
| GNN layers $L_{gnn}$ | {1,2,4,8,16} | {1,2,4,8,16} | {1,2,4,8,16} | {1,2,4,8,16} |
| Feature dimension $d$ | {10, 20, 30} | {64, 128,256} | {64, 128, 256} | {64, 128, 256} |
| Step size $\epsilon$ | {0.001, 0.1, 1.0} | {0.001, 0.1, 1.0} | {0.001, 0.1, 1.0} | {0.001, 0.1, 1.0} |
| Learning rate | {1e-3, 1e-4} | {1e-3, 1e-4} | {1e-3, 1e-4} | {1e-3, 1e-4} |
| Weight decay | {0,1e-6, 1e-5} | {0, 1e-6, 1e-5} | {0, 1e-6, 1e-5} | {0, 1e-6, 1e-5} |
| Activation function ($\sigma$) | ReLU | ELU, GELU, ReLU | ELU, GELU, ReLU | ELU, GELU, ReLU |
| Batch size | {32,64,128} | {32,64,128} | {32, 64, 128} | N/A |

# E ADDITIONAL RESULTS AND COMPARISONS

## E.1 HETEROPHILIC NODE CLASSIFICATION

We report and compare the performance of our TANGO with other recent benchmarks on the heterophilic node classification datasets from Platonov et al. (2023), in Table 9. As can be seen from the Table, TANGO offers strong performance that is similar or better than recent state-of-the-art methods, further demonstrating its effectiveness.

## E.2 ABLATION ON DEPTH: NUMBER OF LAYERS

**Setup.** We study the effect of depth by varying the number of layers and measuring downstream performance on ROMAN-EMPIRE. All runs use identical training settings and data splits; only the depth differs.

**Results.** Table 8 shows that TANGO benefits from increased depth up to a task-dependent plateau. For TANGO-GatedGCN, performance improves steadily and saturates around 16 layers. For TANGO-GPS, gains persist up to 8 to 16 layers and then flatten. Importantly, we do not observe degradation when adding more layers within the explored range.

Table 8: Ablation on the number of layers for ROMAN-EMPIRE. Values are mean classification accuracy (%) $\pm$ standard deviation.

| Layers | 2 | 4 | 8 | 16 | 32 |
|---|---|---|---|---|---|
| TANGO-GatedGCN | $87.13 \pm 0.36$ | $89.08 \pm 0.41$ | $90.80 \pm 0.37$ | $91.89 \pm 0.30$ | $91.82 \pm 0.44$ |
| TANGO-GPS | $86.98 \pm 0.48$ | $88.71 \pm 0.59$ | $91.08 \pm 0.57$ | $91.01 \pm 0.64$ | $91.05 \pm 0.60$ |

## E.3 ADDITIONAL COMPARISONS

The comparisons made in Section 5 offer a focused comparison with directly related methods as well as baseline backbones. In addition to that, we now provide a more comprehensive comparison in Table 12 and Table 13, to further facilitate a comprehensive comparison with recent methods. As can be seen, also under these comparisons, our TANGO offers strong performance.

## E.4 ABLATION STUDY

**Setup.** We conduct two key ablation studies to better understand the contributions of the energy function and the tangential flow in TANGO. Specifically, we aim to answer the following questions:

(i) *Does downstream performance benefit from incorporating a tangential term even when the underlying GNN is not the gradient of an energy function?*

(ii) *Is the observed improvement due to the tangential nature of the added component, or simply due to additional parameters and network?*

To address these questions, we design two controlled experiments. For comprehensive coverage, we evaluate one representative dataset from each benchmark group: ZINC-12k, Roman-empire, Peptides-func, and Diameter. All experiments are run with two backbone architectures, GatedGCN and GPS. For reference, we also report the performance of the original backbones.

**Results.** For ablation (i), we compare TANGO against a variant we call TANGO-NON-ENERGY, in which the gradient-based energy descent term $\nabla_{\mathbf{H}} V_G(\mathbf{H}^{(\ell)})$ in Equation (4) is replaced by intermediate node features from the same GNN backbone, as detailed in Equation (5). These features are computed using the same architecture but are not guaranteed to correspond to the gradient of any scalar energy function. This setup ensures fairness in capacity while removing the energy-based structure. As shown in Table 10, although both variants benefit from the inclusion of the tangential component, the full TANGO consistently outperforms TANGO-NON-ENERGY, confirming that leveraging a valid energy gradient contributes meaningfully to downstream performance.

Table 9: Mean test set score and std averaged over the splits from Platonov et al. (2023). **First**, **second**, and **third** best results for each task are color-coded. We mark each method once – if two variants are among the leading methods, we mark the best-performing variant.

| Model | Roman-empire Acc ↑ | Amazon-ratings Acc ↑ | Minesweeper AUC ↑ | Tolokers AUC ↑ | Questions AUC ↑ |
|---|---|---|---|---|---|
| **MPNNs** | | | | | |
| GIN | $72.82_{\pm0.58}$ | $46.96_{\pm0.44}$ | $88.04_{\pm0.78}$ | $81.79_{\pm0.55}$ | $75.90_{\pm1.03}$ |
| GAT | $80.87_{\pm0.30}$ | $49.09_{\pm0.63}$ | $92.01_{\pm0.68}$ | $83.70_{\pm0.47}$ | $77.43_{\pm1.20}$ |
| GAT-sep | $88.75_{\pm0.41}$ | $52.70_{\pm0.62}$ | $\mathbf{93.91}_{\pm0.35}$ | $83.78_{\pm0.43}$ | $76.79_{\pm0.71}$ |
| Gated-GCN | $74.46_{\pm0.54}$ | $43.00_{\pm0.32}$ | $87.54_{\pm1.22}$ | $77.31_{\pm1.14}$ | $76.61_{\pm1.13}$ |
| GCN | $73.69_{\pm0.74}$ | $48.70_{\pm0.63}$ | $89.75_{\pm0.52}$ | $83.64_{\pm0.67}$ | $76.09_{\pm1.27}$ |
| CO-GNN($\Sigma$, $\Sigma$) | $\mathbf{91.57}_{\pm0.32}$ | $51.28_{\pm0.56}$ | $95.09_{\pm1.18}$ | $83.36_{\pm0.89}$ | $\mathbf{80.02}_{\pm0.86}$ |
| CO-GNN($\mu$, $\mu$) | $91.37_{\pm0.35}$ | $\mathbf{54.17}_{\pm0.37}$ | $\mathbf{97.31}_{\pm0.41}$ | $\mathbf{84.45}_{\pm1.17}$ | $76.54_{\pm0.95}$ |
| SAGE | $85.74_{\pm0.67}$ | $\mathbf{53.63}_{\pm0.39}$ | $93.51_{\pm0.57}$ | $82.43_{\pm0.44}$ | $76.44_{\pm0.62}$ |
| **Graph Transformers** | | | | | |
| Exphormer | $\mathbf{89.03}_{\pm0.37}$ | $53.51_{\pm0.46}$ | $90.74_{\pm0.53}$ | $83.77_{\pm0.78}$ | $73.94_{\pm1.06}$ |
| NAGphormer | $74.34_{\pm0.77}$ | $51.26_{\pm0.72}$ | $84.19_{\pm0.66}$ | $78.32_{\pm0.95}$ | $68.17_{\pm1.53}$ |
| GOAT | $71.59_{\pm1.25}$ | $44.61_{\pm0.50}$ | $81.09_{\pm1.02}$ | $83.11_{\pm1.04}$ | $75.76_{\pm1.66}$ |
| GPS$_{\text{GAT+Performer}}$ (RWSE) | $87.04_{\pm0.58}$ | $49.92_{\pm0.68}$ | $91.08_{\pm0.58}$ | $\mathbf{84.38}_{\pm0.91}$ | $77.14_{\pm1.49}$ |
| GT | $86.51_{\pm0.73}$ | $51.17_{\pm0.66}$ | $91.85_{\pm0.76}$ | $83.23_{\pm0.64}$ | $77.95_{\pm0.68}$ |
| GT-sep | $87.32_{\pm0.39}$ | $52.18_{\pm0.80}$ | $92.29_{\pm0.47}$ | $82.52_{\pm0.92}$ | $78.05_{\pm0.93}$ |
| **Heterophily-Designated GNNs** | | | | | |
| FAGCN | $65.22_{\pm0.56}$ | $44.12_{\pm0.30}$ | $88.17_{\pm0.73}$ | $77.75_{\pm1.05}$ | $77.24_{\pm1.26}$ |
| FSGNN | $79.92_{\pm0.56}$ | $52.74_{\pm0.83}$ | $90.08_{\pm0.70}$ | $82.76_{\pm0.61}$ | $\mathbf{78.86}_{\pm0.92}$ |
| GBK-GNN | $74.57_{\pm0.47}$ | $45.98_{\pm0.71}$ | $90.85_{\pm0.58}$ | $81.01_{\pm0.67}$ | $74.47_{\pm0.86}$ |
| GloGNN | $59.63_{\pm0.69}$ | $36.89_{\pm0.14}$ | $51.08_{\pm1.23}$ | $73.39_{\pm1.17}$ | $65.74_{\pm1.19}$ |
| GPR-GNN | $64.85_{\pm0.27}$ | $44.88_{\pm0.34}$ | $86.24_{\pm0.61}$ | $72.94_{\pm0.97}$ | $55.48_{\pm0.91}$ |
| JacobiConv | $71.14_{\pm0.42}$ | $43.55_{\pm0.48}$ | $89.66_{\pm0.40}$ | $68.66_{\pm0.65}$ | $73.88_{\pm1.16}$ |
| **Ours** | | | | | |
| TANGO$_{\text{GCN}}$ | $89.67_{\pm0.68}$ | $52.98_{\pm0.71}$ | $98.37_{\pm0.49}$ | $85.57_{\pm0.73}$ | $79.86_{\pm1.14}$ |
| TANGO$_{\text{GIN}}$ | $89.19_{\pm0.62}$ | $50.76_{\pm0.47}$ | $97.38_{\pm0.50}$ | $84.39_{\pm0.61}$ | $78.84_{\pm0.96}$ |
| TANGO$_{\text{GatedGCN}}$ | $\mathbf{91.89}_{\pm0.30}$ | $52.60_{\pm0.53}$ | $98.32_{\pm0.59}$ | $85.51_{\pm0.98}$ | $80.39_{\pm1.04}$ |
| TANGO$_{\text{GPS}}$ | $91.08_{\pm0.57}$ | $\mathbf{53.83}_{\pm0.32}$ | $\mathbf{98.39}_{\pm0.54}$ | $\mathbf{85.66}_{\pm1.01}$ | $\mathbf{80.32}_{\pm1.07}$ |

Table 10: Ablation study on the importance of using a gradient of an energy term in Equation (4).

| Model | ZINC-12k MAE ↓ | Roman-empire Acc. ↑ | Peptides-func AP ↑ | Diameter $\log_{10}$(MSE) ↓ |
|---|---|---|---|---|
| GatedGCN | $0.282_{\pm0.015}$ | $74.46_{\pm0.54}$ | $58.64_{\pm0.77}$ | $0.1348_{\pm0.0397}$ |
| TANGO-NON-ENERGY$_{\text{GatedGCN}}$ | $0.138_{\pm0.014}$ | $86.94_{\pm0.43}$ | $68.07_{\pm0.45}$ | $-0.5992_{\pm0.0831}$ |
| TANGO$_{\text{GatedGCN}}$ | $\mathbf{0.128}_{\pm0.011}$ | $\mathbf{91.89}_{\pm0.30}$ | $\mathbf{68.92}_{\pm0.40}$ | $\mathbf{-0.6681}_{\pm0.0745}$ |
| GPS | $0.070_{\pm0.004}$ | $87.04_{\pm0.58}$ | $65.35_{\pm0.41}$ | $-0.5121_{\pm0.0426}$ |
| TANGO-NON-ENERGY$_{\text{GPS}}$ | $0.067_{\pm0.004}$ | $89.00_{\pm0.61}$ | $67.58_{\pm0.39}$ | $-0.7178_{\pm0.0729}$ |
| TANGO$_{\text{GPS}}$ | $\mathbf{0.062}_{\pm0.005}$ | $\mathbf{91.08}_{\pm0.57}$ | $\mathbf{70.21}_{\pm0.43}$ | $\mathbf{-0.9772}_{\pm0.0518}$ |

For ablation (ii), we isolate the effect of the tangential nature of the added direction. In this variant, denoted TANGO-NON-TANGENT, we use the same output from the tangential network as in Equation (9) but omit the orthogonal projection step defined in Equation (3). Thus, while we still introduce an additional GNN term into the dynamics, it is not explicitly orthogonal to the energy gradient. Our results in Table 11 show that while this variant improves the performance compared with the baseline backbone, it also results in a drop in performance compared to the full TANGO. This highlights the importance of the tangential constraint, and its contribution towards improving the utilization of the learned energy function, as discussed in Section 4. Together, these ablations underscore the importance of both components in our design: (i) the principled learned energy descent, and (ii) the structured tangential update, as crucial for effective and flexible feature evolution.

Table 11: The importance of using a tangential term to the energy term in Equation (4).

| Model | ZINC-12k MAE ↓ | Roman-empire Acc. ↑ | Peptides-func AP ↑ | Diameter $\log_{10}$(MSE) ↓ |
|---|---|---|---|---|
| GatedGCN | $0.282_{\pm 0.015}$ | $74.46_{\pm 0.54}$ | $58.64_{\pm 0.77}$ | $0.1348_{\pm 0.0397}$ |
| TANGO-NON-TANGENT$_{GatedGCN}$ | $0.186_{\pm 0.016}$ | $83.59_{\pm 0.48}$ | $68.01_{\pm 0.52}$ | $-0.2193_{\pm 0.0899}$ |
| TANGO$_{GatedGCN}$ | $\mathbf{0.128}_{\pm 0.011}$ | $\mathbf{91.89}_{\pm 0.30}$ | $\mathbf{68.92}_{\pm 0.40}$ | $\mathbf{-0.6681}_{\pm 0.0745}$ |
| GPS | $0.070_{\pm 0.004}$ | $87.04_{\pm 0.58}$ | $65.35_{\pm 0.41}$ | $-0.5121_{\pm 0.0426}$ |
| TANGO-NON-TANGENT$_{GPS}$ | $0.066_{\pm 0.010}$ | $88.57_{\pm 0.72}$ | $67.33_{\pm 0.59}$ | $-0.2916_{\pm 0.0404}$ |
| TANGO$_{GPS}$ | $\mathbf{0.062}_{\pm 0.005}$ | $\mathbf{91.08}_{\pm 0.57}$ | $\mathbf{70.21}_{\pm 0.43}$ | $\mathbf{-0.9772}_{\pm 0.0518}$ |

Table 12: Results for Peptides-func and Peptides-struct averaged over 3 training seeds. Baseline results are taken from Dwivedi et al. (2022b) and Gutteridge et al. (2023). Re-evaluated methods employ the 3-layer MLP readout proposed in Tönshoff et al. (2023a). Note that all MPNN-based methods include structural and positional encoding. ‡ means 3-layer MLP readout and residual connections are employed based on (Tönshoff et al., 2023a). This table is an extended version of the focused Table 3.

| Model | Peptides-func AP ↑ | Peptides-struct MAE ↓ |
|---|---|---|
| **MPNNs** | | |
| GCN | $59.30_{\pm 0.23}$ | $0.3496_{\pm 0.0013}$ |
| GINE | $54.98_{\pm 0.79}$ | $0.3547_{\pm 0.0045}$ |
| GCNII | $55.43_{\pm 0.78}$ | $0.3471_{\pm 0.0010}$ |
| GatedGCN | $58.64_{\pm 0.77}$ | $0.3420_{\pm 0.0013}$ |
| **Multi-hop GNNs** | | |
| DIGL+MPNN | $64.69_{\pm 0.19}$ | $0.3173_{\pm 0.0007}$ |
| DIGL+MPNN+LapPE | $68.30_{\pm 0.26}$ | $0.2616_{\pm 0.0018}$ |
| MixHop-GCN | $65.92_{\pm 0.36}$ | $0.2921_{\pm 0.0023}$ |
| MixHop-GCN+LapPE | $68.43_{\pm 0.49}$ | $0.2614_{\pm 0.0023}$ |
| DRew-GCN | $69.96_{\pm 0.76}$ | $0.2781_{\pm 0.0028}$ |
| DRew-GCN+LapPE | $71.50_{\pm 0.44}$ | $0.2536_{\pm 0.0015}$ |
| DRew-GIN | $69.40_{\pm 0.74}$ | $0.2799_{\pm 0.0016}$ |
| DRew-GIN+LapPE | $71.26_{\pm 0.45}$ | $0.2606_{\pm 0.0014}$ |
| DRew-GatedGCN | $67.33_{\pm 0.94}$ | $0.2699_{\pm 0.0018}$ |
| DRew-GatedGCN+LapPE | $69.77_{\pm 0.26}$ | $0.2539_{\pm 0.0007}$ |
| **Transformers** | | |
| Transformer+LapPE | $63.26_{\pm 1.26}$ | $0.2529_{\pm 0.0016}$ |
| SAN+LapPE | $63.84_{\pm 1.21}$ | $0.2683_{\pm 0.0043}$ |
| GraphGPS+LapPE | $65.35_{\pm 0.41}$ | $0.2500_{\pm 0.0005}$ |
| **Modified and Re-evaluated‡** | | |
| GCN | $68.60_{\pm 0.50}$ | $0.2460_{\pm 0.0007}$ |
| GINE | $66.21_{\pm 0.67}$ | $0.2473_{\pm 0.0017}$ |
| GatedGCN | $67.65_{\pm 0.47}$ | $0.2477_{\pm 0.0009}$ |
| GraphGPS | $65.34_{\pm 0.91}$ | $0.2509_{\pm 0.0014}$ |
| **DE-GNNs** | | |
| GRAND | $57.89_{\pm 0.62}$ | $0.3418_{\pm 0.0015}$ |
| GraphCON | $60.22_{\pm 0.68}$ | $0.2778_{\pm 0.0018}$ |
| A-DGN | $59.75_{\pm 0.44}$ | $0.2874_{\pm 0.0021}$ |
| SWAN | $67.51_{\pm 0.39}$ | $0.2485_{\pm 0.0009}$ |
| **Graph SSMs** | | |
| Graph-Mamba | $67.39_{\pm 0.87}$ | $0.2478_{\pm 0.0016}$ |
| GMN | $70.71_{\pm 0.83}$ | $0.2473_{\pm 0.0025}$ |
| **Ours** | | |
| TANGO$_{GCN}$ | $69.17_{\pm 0.31}$ | $0.2432_{\pm 0.0011}$ |
| TANGO$_{GIN}$ | $68.78_{\pm 0.66}$ | $0.2440_{\pm 0.0024}$ |
| TANGO$_{GATEDGCN}$ | $68.92_{\pm 0.40}$ | $0.2451_{\pm 0.0006}$ |
| TANGO$_{GPS}$ | $70.21_{\pm 0.43}$ | $0.2422_{\pm 0.0014}$ |

Table 13: Mean test set score and std averaged over the splits from Platonov et al. (2023). This table is an extended version of the focused Table 9. Baseline results are reported from Finkelshtein et al. (2024); Platonov et al. (2023); Müller et al. (2024); Luan et al. (2024).

| Model | Roman-empire Acc ↑ | Amazon-ratings Acc ↑ | Minesweeper AUC ↑ | Tolokers AUC ↑ | Questions AUC ↑ |
|---|---|---|---|---|---|
| **MPNNs** | | | | | |
| GIN | $72.82_{\pm0.58}$ | $46.96_{\pm0.44}$ | $88.04_{\pm0.78}$ | $81.79_{\pm0.55}$ | $75.90_{\pm1.03}$ |
| GAT | $80.87_{\pm0.30}$ | $49.09_{\pm0.63}$ | $92.01_{\pm0.68}$ | $83.70_{\pm0.47}$ | $77.43_{\pm1.20}$ |
| GAT-sep | $88.75_{\pm0.41}$ | $52.70_{\pm0.62}$ | $93.91_{\pm0.35}$ | $83.78_{\pm0.43}$ | $76.79_{\pm0.71}$ |
| GAT (LapPE) | $84.80_{\pm0.46}$ | $44.90_{\pm0.73}$ | $93.50_{\pm0.54}$ | $84.99_{\pm0.54}$ | $76.55_{\pm0.84}$ |
| GAT (RWSE) | $86.62_{\pm0.53}$ | $48.58_{\pm0.41}$ | $92.53_{\pm0.65}$ | $85.02_{\pm0.67}$ | $77.83_{\pm1.22}$ |
| GAT (DEG) | $85.51_{\pm0.56}$ | $51.65_{\pm0.60}$ | $93.04_{\pm0.62}$ | $84.22_{\pm0.81}$ | $77.10_{\pm1.23}$ |
| Gated-GCN | $74.46_{\pm0.54}$ | $43.00_{\pm0.32}$ | $87.54_{\pm1.22}$ | $77.31_{\pm1.14}$ | $76.61_{\pm1.13}$ |
| GCN | $73.69_{\pm0.74}$ | $48.70_{\pm0.63}$ | $89.75_{\pm0.52}$ | $83.64_{\pm0.67}$ | $76.09_{\pm1.27}$ |
| GCN (LapPE) | $83.37_{\pm0.55}$ | $44.35_{\pm0.36}$ | $94.26_{\pm0.49}$ | $84.95_{\pm0.78}$ | $77.79_{\pm1.34}$ |
| GCN (RWSE) | $84.84_{\pm0.55}$ | $46.40_{\pm0.55}$ | $93.84_{\pm0.48}$ | $85.11_{\pm0.77}$ | $77.81_{\pm1.40}$ |
| GCN (DEG) | $84.21_{\pm0.47}$ | $50.01_{\pm0.69}$ | $94.14_{\pm0.50}$ | $82.51_{\pm0.83}$ | $76.96_{\pm1.21}$ |
| CO-GNN$(\Sigma, \Sigma)$ | $91.57_{\pm0.32}$ | $51.28_{\pm0.56}$ | $95.09_{\pm1.18}$ | $83.36_{\pm0.89}$ | $80.02_{\pm0.86}$ |
| CO-GNN$(\mu, \mu)$ | $91.37_{\pm0.35}$ | $54.17_{\pm0.37}$ | $97.31_{\pm0.41}$ | $84.45_{\pm1.17}$ | $76.54_{\pm0.95}$ |
| SAGE | $85.74_{\pm0.67}$ | $53.63_{\pm0.39}$ | $93.51_{\pm0.57}$ | $82.43_{\pm0.44}$ | $76.44_{\pm0.62}$ |
| **Graph Transformers** | | | | | |
| Exphormer | $89.03_{\pm0.37}$ | $53.51_{\pm0.46}$ | $90.74_{\pm0.53}$ | $83.77_{\pm0.78}$ | $73.94_{\pm1.06}$ |
| NAGphormer | $74.34_{\pm0.77}$ | $51.26_{\pm0.72}$ | $84.19_{\pm0.66}$ | $78.32_{\pm0.95}$ | $68.17_{\pm1.53}$ |
| GOAT | $71.59_{\pm1.25}$ | $44.61_{\pm0.50}$ | $81.09_{\pm1.02}$ | $83.11_{\pm1.04}$ | $75.76_{\pm1.66}$ |
| GPS | $82.00_{\pm0.61}$ | $53.10_{\pm0.42}$ | $90.63_{\pm0.67}$ | $83.71_{\pm0.48}$ | $71.73_{\pm1.47}$ |
| GPS$_{\text{GCN+Performer}}$ (LapPE) | $83.96_{\pm0.53}$ | $48.20_{\pm0.67}$ | $93.85_{\pm0.41}$ | $84.72_{\pm0.77}$ | $77.85_{\pm1.25}$ |
| GPS$_{\text{GCN+Performer}}$ (RWSE) | $84.72_{\pm0.65}$ | $48.08_{\pm0.85}$ | $92.88_{\pm0.50}$ | $84.81_{\pm0.86}$ | $76.45_{\pm1.51}$ |
| GPS$_{\text{GCN+Performer}}$ (DEG) | $83.38_{\pm0.68}$ | $48.93_{\pm0.47}$ | $93.60_{\pm0.47}$ | $80.49_{\pm0.97}$ | $74.24_{\pm1.18}$ |
| GPS$_{\text{GAT+Performer}}$ (LapPE) | $85.93_{\pm0.52}$ | $48.86_{\pm0.38}$ | $92.62_{\pm0.79}$ | $84.62_{\pm0.54}$ | $76.71_{\pm0.98}$ |
| GPS$_{\text{GAT+Performer}}$ (RWSE) | $87.04_{\pm0.58}$ | $49.92_{\pm0.68}$ | $91.08_{\pm0.58}$ | $84.38_{\pm0.91}$ | $77.14_{\pm1.49}$ |
| GPS$_{\text{GAT+Performer}}$ (DEG) | $85.54_{\pm0.58}$ | $51.03_{\pm0.60}$ | $91.52_{\pm0.46}$ | $82.45_{\pm0.89}$ | $76.51_{\pm1.19}$ |
| GPS$_{\text{GCN+Transformer}}$ (LapPE) | OOM | OOM | $91.82_{\pm0.41}$ | $83.51_{\pm0.93}$ | OOM |
| GPS$_{\text{GCN+Transformer}}$ (RWSE) | OOM | OOM | $91.17_{\pm0.51}$ | $83.53_{\pm1.06}$ | OOM |
| GPS$_{\text{GCN+Transformer}}$ (DEG) | OOM | OOM | $91.76_{\pm0.61}$ | $80.82_{\pm0.95}$ | OOM |
| GPS$_{\text{GAT+Transformer}}$ (LapPE) | OOM | OOM | $92.29_{\pm0.61}$ | $84.70_{\pm0.56}$ | OOM |
| GPS$_{\text{GAT+Transformer}}$ (RWSE) | OOM | OOM | $90.82_{\pm0.56}$ | $84.01_{\pm0.96}$ | OOM |
| GPS$_{\text{GAT+Transformer}}$ (DEG) | OOM | OOM | $91.58_{\pm0.56}$ | $81.89_{\pm0.85}$ | OOM |
| GT | $86.51_{\pm0.73}$ | $51.17_{\pm0.66}$ | $91.85_{\pm0.76}$ | $83.23_{\pm0.64}$ | $77.95_{\pm0.68}$ |
| GT-sep | $87.32_{\pm0.39}$ | $52.18_{\pm0.80}$ | $92.29_{\pm0.47}$ | $82.52_{\pm0.92}$ | $78.05_{\pm0.93}$ |
| **Heterophily-Designated GNNs** | | | | | |
| CPGNN | $63.96_{\pm0.62}$ | $39.79_{\pm0.77}$ | $52.03_{\pm5.46}$ | $73.36_{\pm1.01}$ | $65.96_{\pm1.95}$ |
| FAGCN | $65.22_{\pm0.56}$ | $44.12_{\pm0.30}$ | $88.17_{\pm0.73}$ | $77.75_{\pm1.05}$ | $77.24_{\pm1.26}$ |
| FSGNN | $79.92_{\pm0.56}$ | $52.74_{\pm0.83}$ | $90.08_{\pm0.70}$ | $82.76_{\pm0.61}$ | $78.86_{\pm0.92}$ |
| GBK-GNN | $74.57_{\pm0.47}$ | $45.98_{\pm0.71}$ | $90.85_{\pm0.58}$ | $81.01_{\pm0.67}$ | $74.47_{\pm0.86}$ |
| GloGNN | $59.63_{\pm0.69}$ | $36.89_{\pm0.14}$ | $51.08_{\pm1.23}$ | $73.39_{\pm1.17}$ | $65.74_{\pm1.19}$ |
| GPR-GNN | $64.85_{\pm0.27}$ | $44.88_{\pm0.34}$ | $86.24_{\pm0.61}$ | $72.94_{\pm0.97}$ | $55.48_{\pm0.91}$ |
| H2GCN | $60.11_{\pm0.52}$ | $36.47_{\pm0.23}$ | $89.71_{\pm0.31}$ | $73.35_{\pm1.01}$ | $63.59_{\pm1.46}$ |
| JacobiConv | $71.14_{\pm0.42}$ | $43.55_{\pm0.48}$ | $89.66_{\pm0.40}$ | $68.66_{\pm0.65}$ | $73.88_{\pm1.16}$ |
| **Graph SSMs** | | | | | |
| GMN | $87.69_{\pm0.50}$ | $54.07_{\pm0.31}$ | $91.01_{\pm0.23}$ | $84.52_{\pm0.21}$ | – |
| GPS + Mamba | $83.10_{\pm0.28}$ | $45.13_{\pm0.97}$ | $89.93_{\pm0.54}$ | $83.70_{\pm1.05}$ | – |
| **Ours** | | | | | |
| TANGO$_{\text{GCN}}$ | $89.67_{\pm0.68}$ | $52.98_{\pm0.71}$ | $98.37_{\pm0.49}$ | $85.57_{\pm0.73}$ | $79.86_{\pm1.14}$ |
| TANGO$_{\text{GIN}}$ | $89.19_{\pm0.62}$ | $50.76_{\pm0.47}$ | $97.38_{\pm0.50}$ | $84.39_{\pm0.61}$ | $78.84_{\pm0.96}$ |
| TANGO$_{\text{GatedGCN}}$ | $91.89_{\pm0.30}$ | $52.60_{\pm0.53}$ | $98.32_{\pm0.59}$ | $85.51_{\pm0.98}$ | $80.39_{\pm1.04}$ |
| TANGO$_{\text{GPS}}$ | $91.08_{\pm0.57}$ | $53.83_{\pm0.32}$ | $98.39_{\pm0.54}$ | $85.66_{\pm1.01}$ | $80.32_{\pm1.07}$ |

