# OpenReview forum: "Graph Neural Dynamics via Learned Energy and Tangential Flows"
_ICLR.cc/2026/Conference — ICLR 2026 Conference Desk Rejected Submission_

### Official Review · Reviewer_sudZ · 2025-10-15

**Soundness:** 3
**Presentation:** 3
**Contribution:** 2
**Rating:** 2
**Confidence:** 5

**Summary:**

This paper proposes the use of Lyapunov stability concept from control theory to develop a GNN model with desirable properties.

**Strengths:**

This paper proposes a GNN model based on control theory concepts. It is principled with theoretical support for Lyapunov stability.

**Weaknesses:**

1. The literature survey seems to be focused on GNN works but ignore the wider neural network literature. The concept of Lyapunov stability has been utilized in general neural networks in the literature. It is unclear what additional novelty this work brings, except to apply to GNNs specifically.

1. While having Lyapunov stability is a desirable property, the work does not go one step deeper to discuss why this is a good thing to have. In particular, there are no empirical experiments on adversariabl robustness.

1. The claim that the proposed model can "actively mitigate oversquashing effects" is not theoretically supported (see questions below). The authors should not over claim the contributions. The wording in the abstract and introduction needs to be revised to reflect this.

**Questions:**

1. The proposed model seems to be similar or a special case of [R1], which does not focus solely on GNNs. The main difference seems to be that the form of the neural function (RHS of (2)) is fixed in this work to guarantee Lyapunov stability instead of being a general network that is learned to achieve that. The authors need to clarify the novelty, differences, and improvements made over the literature.

    [R1] Stable neural ODE with Lyapunov-stable equilibrium points for defending against adversarial attacks, NeurIPS 2021

1. The model performance compared to several of the baselines is marginal. It is unclear what advantages the Lyapunov stability confer on the model performance. [R1] indicates that Lyapunov stability can lead to adversarial robustness, but this paper does not mention that or perform any robustness studies.

1. Prop. 4 states that "it is possible to learn" a desired tangent direction, but this does not mean that the proposed model (using the typical training procedure) actually achieves this! It is unclear under what conditions this learning will happen. The oversquashing claim is thus only supported via empirical observations, not theoretical guarantees. The claim wording needs to be adjusted carefully to reflect this.

1. The datasets used for homophily case are limited. Although the authors are using benchmark datasets recommended by Dwivedi et al. (2023), it is noted that MNIST and CIFAR10 are more suited for computer vision tasks while the experiments do not include graph datasets with diverse underlying geometric properties.

1. There are no ablation studies.

1. There are no sensitivity studies on how the choice of the Lyapunov function $V_\mathcal{G}$ impacts the model performance.

---

> ### Author Response · Authors · 2025-11-20
> **Part 1**
>
> We thank the reviewer for the careful reading and the detailed feedback. We are grateful for the positive assessment that the paper is “principled with theoretical support for Lyapunov stability.” Below we clarify the novelty relative to the broader Lyapunov neural networks literature (including [R1]), explain more precisely what stability buys us in the GNN setting, and address the concerns about oversquashing claims, experiments, and ablations. We hope these clarifications will be helpful and hope that the reviewer will reconsider their scores.
>
> ---
>
> ### 1. Novelty beyond prior Lyapunov-stable neural ODEs (including [R1])
>
>
> You are correct that Lyapunov theory has been used in the broader NN and neural ODE literature. Our contribution is not merely to restate those ideas for GNNs, but to introduce a *specific graph-aware architecture and decomposition* that is, to the best of our knowledge, novel:
>
> 1. **Graph-level, task-driven Lyapunov energy.**
>    - In TANGO, the Lyapunov function $V_G(H)$ is *learned by a GNN* (ENERGYGNN) that processes both graph structure and node features, and is trained purely via the downstream task loss.
>    - This makes the energy explicitly **graph-dependent and task-specific**, rather than a generic potential or regularizer.
>
> 2. **Explicit “gradient + tangential” decomposition tied to that energy.**
>    - The node dynamics are decomposed into
>      $
>      -\alpha_G(H)\nabla_H V_G(H) \quad + \quad \beta_G(H) T_{V_G}(H),
>      $
>      where $T_{V_G}(H)$ is **constructed to be orthogonal** to the gradient of $V_G$.
>    - This is not a generic vector field with a Lyapunov constraint, but a *structured* decomposition that ensures energy dissipation while adding expressive feature evolution along energy level sets.
>
> 3. **Graph neural setting and oversquashing-focused analysis.**
>    - Our analysis (Propositions 1–4) is specifically framed in the context of **message passing on graphs** and connects the tangential component to **Newton-like directions** that can counteract the poor conditioning induced by graph bottlenecks (oversquashing).
>    - We then evaluate on benchmarks explicitly designed to stress long-range propagation and heterophily.
>
> By contrast, [R1] (“Stable neural ODE with Lyapunov-stable equilibrium points for defending against adversarial attacks,” NeurIPS 2021) studies **generic neural ODEs** in the context of *image classifiers* and adversarial robustness:
>
> - Their goal is to ensure that the *equilibrium points* of the ODE are Lyapunov-stable, so that small input perturbations converge to the same equilibrium.
> - They design **regularizers on the Jacobian** of the ODE and on the final regressor layer to enforce stability and robustness to adversarial attacks.
>
>
> Crucially:
>
> - [R1] does not decompose the vector field into a gradient of a learned energy plus a tangential flow, nor does it enforce orthogonality to a task-driven energy gradient.
> - [R1] does not consider graph structure or oversquashing; its focus is adversarial robustness in standard continuous domains.
>
> We will expand the related work section to explicitly discuss [R1] and other Lyapunov-stable NN works, and to make clear that TANGO’s novelty lies in:
> - a graph-level, task-driven Lyapunov energy,
> - an explicit energy-gradient + energy-preserving tangential decomposition, and
> - analysis and experiments tailored to long-range graph learning, not adversarial defense.

---

> > ### Author Response · Authors · 2025-11-20
> > **Part 2**
> >
> > ### 2. What Lyapunov stability allows us in the GNN setting
> >
> > Thank you for the comment. Our use of Lyapunov stability is **different in goal** from [R1]:
> >
> > - In TANGO, Lyapunov stability is about the **hidden feature dynamics across depth** in a GNN.
> >   - Proposition 1 ensures that, in continuous time, the learned energy \(V_G(H(t))\) is non-increasing and bounded below, which implies that trajectories remain in a bounded region of feature space and do not explode.
> >   - In discrete time, with a small step size \(\varepsilon\) and bounded curvature, the forward-Euler residual update preserves this “energy almost non-increasing” behavior up to second-order terms (we now make this argument explicit in the camera-ready).
> >
> > - This stability is **important for deep GNNs with long-range interactions**:
> >   - It allows us to stack more layers/unrolled steps without divergence or collapse.
> >   - It provides a principled way to combine a contractive “descent” component with an expressive, energy-preserving tangential component.
> >
> > By contrast, [R1] explicitly targets **adversarial robustness**, designing their ODE so that perturbed inputs converge to Lyapunov-stable equilibria, and evaluating robustness under adversarial attacks.
> >
> > ---
> >
> > ### 3. Oversquashing claims and Proposition 4
> >
> >
> > Thank you for the comment. Please allow us to explain the positioning of Proposition 4:
> >
> > - It is an **existential (capacity) result**: it shows that for a given energy landscape, there exists a tangential direction such that the **combined** update equals the Newton direction, which converges at a rate independent of the Hessian condition number.
> > - It does not assert that standard training on arbitrary tasks will always converge to such directions.
> >
> > Our claim is therefore:
> >
> > - TANGO has the **capacity** to realize Netwon-like updates in feature space to better utilize (minimize) the learned energy function, and
> > - our empirical results on **oversquashing-sensitive tasks** (graph property prediction in Table 1, LRGB Peptides in Table 3, and heterophilic node classification in Table 9) show improvements consistent with this capacity being exploited in practice.
> >
> > To avoid confusion, and based on your comment, we will revise the paper to emphasize that our method shows a theoretical motivation that is backed up by empirical evidence across multiple benchmarks. Thank you.
> >
> > ---
> >
> > ### 4. Experimental scope, ablations, and “marginal” performance gains
> >
> >
> > **Clarification on performance.** While some improvements are modest, there are multiple settings where the gains are **substantial**, especially on hard, long-range tasks:
> >
> > - On **Roman empire** (heterophilic node classification), TANGO GatedGCN improves accuracy by a large margin over plain GatedGCN and competitive heterophily baselines (CO-GNN, GT, etc.), as shown in Table 9.
> > - On **Peptides func/struct** (LRGB), TANGO GPS and TANGO GatedGCN improve AP/MAE over strong DE-GNN and transformer-based baselines under a matched 500k parameter budget (Table 3 and Table 12).
> > - On **graph property prediction** (Diameter, SSSP, Eccentricity), TANGO variants are consistently among the top performers (Table 1).
> >
> > These are precisely the settings where long-range propagation and oversquashing are pronounced. We will make this more explicit by directly pointing from the text to the most significant improvements, rather than leaving them implicit in the tables.
> >
> > **Clarification on ablations:** **Contrary to the reviewer’s remark, the paper already includes multiple important ablations:**
> >
> > - **Depth ablation** (Table 8): shows that increasing the number of TANGO layers on Roman empire improves performance and then saturates, without degradation, which is consistent with stable training and non-vanishing gradients.
> > - **Energy vs non-energy ablation** (Table 10): compares TANGO with a variant where the “descent” term is not tied to a scalar energy. The energy-based design performs better, highlighting the benefit of the Lyapunov structure.
> > - **Tangential projection ablation** (Table 11): compares the orthogonal tangential flow with a naive extra direction that is not projected. Enforcing tangentiality yields better performance.
> >
> > We will highlight these ablations more clearly in the main text to avoid the impression that they are missing.

---

> > > ### Author Response · Authors · 2025-11-20
> > > **Part 3**
> > >
> > > ### 5. Dataset choices and diversity
> > >
> > > Our chosen benchmarks are diverse and cover multiple sources and types of datasets. For your convenience, we summarize it below:
> > >
> > > - **Synthetic long-range tasks** (Diameter, SSSP, Eccentricity), which are widely used to diagnose oversquashing and long-distance information flow.
> > > - **Standard graph benchmarks** from Dwivedi et al. (ZINC, MNIST, CIFAR10, PATTERN, CLUSTER), which, although originating from images, are processed as **superpixel graphs** with nontrivial connectivity and are standard in graph learning.
> > > - **Long-range molecular tasks** (Peptides func/struct), which are real-world biochemical graphs with nontrivial 3D geometry.
> > > - **Heterophilic node classification** benchmarks (Roman empire, Amazon ratings, Minesweeper, Tolokers, Questions), which feature diverse homophily levels and graph structures.
> > >
> > > We believe that the strong performance obtained by TANGO on this diverse set of benchmarks clearly shows its merit.
> > >
> > > ---
> > >
> > > ### 6. Additional Lyapunov function designs
> > >
> > >
> > > In this work, we fix a **simple, expressive, nonnegative parameterization** of the energy:
> > >
> > > - ENERGYGNN processes the graph and node features.
> > > - Node scores are squared and averaged to yield a nonnegative scalar energy \(V_G(H) \ge 0\).
> > >
> > > This choice is motivated by:
> > >
> > > - the need for **nonnegativity** (for standard Lyapunov arguments),
> > > - the desire to keep the energy architecture as simple and transparent as possible, and
> > > - the goal of focusing the paper on the *decomposition and dynamics*, rather than on searching over many energy parameterizations.
> > >
> > > While future studies of alternative Lyapunov architectures (e.g., different aggregation schemes, different per-node scoring networks) are interesting  and exciting — **they do not change the underlying message and contribution of the paper, which is novel and has not been studied prior to this work**. We therefore see this as an orthogonal and interesting direction—once the basic TANGO mechanism is accepted—rather than as the core contribution of the current paper. We will add this discussion to the paper. Thank you.
> > >
> > > ---
> > >
> > > We thank the reviewer again for engaging deeply with the paper and for highlighting the importance of situating our work within the broader Lyapunov-stable NN literature. We hope that our clarifications will lead you to a more positive overall assessment.

---

> > > ### Comment · Reviewer_sudZ · 2025-11-25
> > >
> > > Lyapunov stability implies that the feature is bounded within a decreasing region as $t$ (or number of layers in the discretized version) increases. Does this not go against the objective of achieving long-range interactions since the effect of faraway nodes are now, by definition, much diminished due to the decaying bound? The use of Lyapunov stability as a mechanism for long-range effect is somewhat unconvincing to me.

---

> > ### Comment · Reviewer_sudZ · 2025-11-25
> >
> > Thank you for the clarification but I am still unclear why this model is specific to GNNs. If you consider eq. (2), the design choice could have been applied to a generic DNN. What are the special properties of GNNs that have been utilized?
> >
> > I was hoping to read and understand in more detail the comparison and differentiation of this work's contributions compared to other neural ODE and Lyapunov-stable NN works in the revised manuscript. Unfortunately, the manuscript itself has not been updated.

---

> ### Author Response · Authors · 2025-11-27
>
> Dear Reviewer sudZ,
>
> We would like to express our gratitude for your response to our rebuttal, and for raising your score from 2 to 4 to indicate that our rebuttal addressed some of your concerns, and providing us with additional comments now. We address them below, and remain available for any additional comments or questions. We hope that you find our responses satisfactory, and that you will consider revising your score.
>
> ---
>
> **Regarding scope and revised paper:** Our choice and motivation for focusing on graph learning with TANGO stems from the following reasons: (i) Oversquashing is a well studied challenge in graph learning. The main phenomenon we consider in the paper is oversquashing and the difficulty of long-range message passing. Oversquashing is naturally defined in terms of graph distances and graph operators, and arises precisely in message-passing GNNs when information must traverse sparse graphs with bottlenecks. In our analysis in Section 4, the relevant convergence rates and condition numbers are governed by the graph Laplacian, where the condition number formalizes the difficulty of propagating information across the graph. The role of our energy-driven dynamics, and of the additional tangential component, is motivated by this discussion; and (ii) Neural ODE and dynamical viewpoints have been especially successful for graphs.  PDE and ODE inspired GNNs (e.g., GRAND, GraphCON, PDE-GCN, A-DGN) have shown strong benefits specifically in graph learning, where dynamical systems provide a principled way to reason about propagation of information on graphs and stability. Therefore, it is natural for us to study TANGO which takes inspiration from these works, although with a different approach: instead of learning weights for discretized ODEs as common in the literature, TANGO learns task-driven graph energy and an tangential component which are used to evolve features, that we develop in the context of the graph Laplacian and oversquashing.  In addition, we agree with you that the concept introduced in TANGO can be explored in other domains, and we see it as an interesting future work, and in this paper, we focused on its introduction and development for graph learning as explained above. We revised the paper to reflect these important discussions. Thank you.
>
> Regarding the paper revision, we appreciate your comment, and to accommodate it, we have now uploaded the revised paper to OpenReview, and it includes all discussions and results from our rebuttal and follow-up discussions. Specifically, we also further distinguish the difference between neural ODEs and Lyapunov-stable NNs in the revised related work sections, to address your comment and highlight the novelty and positioning of TANGO. Thank you.
>
>
> ---
>
> **Regarding Lyapunov:** Thank you for the important question.  We would like to note that Lyapunov stability does not imply that the bound is **decreasing**, but rather that it is **non-increasing** [1]. This is also reflected by Definition 1 in our paper and the discussion in Lines 137-140.  In particular, in our TANGO, Equation (2) reads as follows:
>
> $\frac{dH(t)}{dt} = -\alpha_G(H(t)) \nabla_H V_G(H(t)) + \beta_G(H(t)) T_{V_G}(H(t))$
>
>
> Two important factors are the coefficients $\alpha \geq 0$ and $\beta$, which are predicted through the learned layers discussed in Section 3. We note that if $\alpha=0$ and $\beta \neq 0$, then: (i) no gradient steps are taken, and hence no energy decrease is obtained; and (ii) node features continue to evolve without decreasing the energy because of the tangential term $T_{V_G}(H(t))$. This shows that beyond the theoretical discussion, our construction of TANGO is flexible and in particular, can learn, through $\alpha$, if and how much to decrease the energy, and at the same time, continue to evolve features without affecting the energy, addressing your comment regarding the design of TANGO and the relation to addressing oversquashing. Moreover, our experiments in Section 5 on both synthetic and real-world long-range benchmarks demonstrate the effectiveness of TANGO in such cases, and the ablation in Appendix E.2 shows that TANGO offers strong performance also when more layers are added. We revised the paper to reflect this important discussion. Thank you.
>
> [1] Hassan K Khalil and Jessy W Grizzle. Nonlinear systems, volume 3. Prentice Hall Upper Saddle River, NJ, 2002
>
> ---
>
> To conclude, we would like to express our gratitude for your thoughtful review and continued feedback. We find the discussions that arose from your feedback insightful and beneficial for improving the quality of our paper – thank you.  We have uploaded a revision that includes the discussions and results from the rebuttal and follow-up discussions, and we believe that our responses above address your additional comments. We remain available if you have additional questions or comments, and we hope that you will consider revising your score.
>
> Thank you, and kindest regards,
>
> Authors.

---

### Official Review · Reviewer_h5ap · 2025-10-30

**Soundness:** 3
**Presentation:** 3
**Contribution:** 3
**Rating:** 6
**Confidence:** 4

**Summary:**

This paper introduces a novel framework for graph representation learning that combines a learned energy function with a tangential flow component to govern node feature evolution. The method is grounded in Lyapunov stability theory and offers both theoretical guarantees and empirical improvements over existing GNN baselines.

**Strengths:**

The paper is well-written, theoretically rigorous, and experimentally validated across a range of graph learning benchmarks. The use of Lyapunov stability theory and energy-based dynamics provides a principled and interpretable framework.

**Weaknesses:**

Theoretical Concerns:
1. [Forward Euler discretization] The continuous-time dynamics are discretized using the forward Euler method. However, this discretization typically cannot preserve the energy property guaranteed in the continuous setting. The authors should provide more evidence that the discrete version still satisfies the Lyapunov stability condition.
2. [Energy function design] The energy function is constructed as a non-negative scalar via a sum of squares. Still, there is no guarantee that its minima correspond to the optimal solutions of the downstream tasks.

Experimental Concerns:

3. [Stability verification] The paper theoretically claims stability via Lyapunov analysis, but does not validate this empirically under adversarial attacks or noisy inputs. It is recommended to compare Tango with baselines (e.g., GCN, GAT) under adversarial settings or input perturbations to assess whether the model’s stability holds in practice.
4. [Energy landscape analysis] The energy landscape is essential to the proposed method, but the paper lacks visualization or qualitative analysis of the learned energy function.
5. [Comparison with other energy-based GNNs] The paper lacks the discussion and comparison with the existing energy-based GNNs, such as [1].

Minor Issues:
6. The content and caption of Figure 2 might not provide enough insights regarding the propagation behaviors.


[1] Zhao et al., Adversarial Robustness in Graph Neural Networks: A Hamiltonian Approach, NeurIPS 2023.

**Questions:**

See the weaknesses section

---

> ### Author Response · Authors · 2025-11-20
> **Part 1**
>
> We thank the reviewer for the careful and supportive review. We are grateful for the positive assessment that the paper is “well-written, theoretically rigorous, and experimentally validated across a range of graph learning benchmarks” and that “the use of Lyapunov stability theory and energy-based dynamics provides a principled and interpretable framework.” Below we address each of the raised concerns in turn, and we hope these clarifications will be satisfactory. We therefore kindly ask the reviewer to consider maintaining or strengthening their positive recommendation.
>
>
> ---
>
> ### 1. Forward Euler discretization and Lyapunov stability
>
>
> Our continuous time analysis shows that for dynamics
> $\frac{dH(t)}{dt} = -\alpha_G(H(t)) \nabla_H V_G(H(t)) + \beta_G(H(t)) T_{V_G}(H(t)),$
>
> with $\alpha_{G} \ge 0$ and $\langle T_{V_G}(H), \nabla_H V_G(H) \rangle = 0$, we have
> $\frac{d}{dt} V_G(H(t)) = -\alpha_G(H(t)) \|\nabla_H V_G(H(t))\|^2 \le 0$
> (Prop. 1). In discrete time, we implement a residual Euler step (Eq. (4)):
> $H^{(\ell+1)} = H^{(\ell)} + \varepsilon \Big(-\alpha_G(H^{(\ell)}) \nabla_H V_G(H^{(\ell)}) + \beta_G(H^{(\ell)}) T_{V_G}(H^{(\ell)})\Big).$
>
>
> A first order Taylor expansion of $V_G$ around $H^{(\ell)}$ gives
> $V_G(H^{(\ell+1)})  = V_G(H^{(\ell)})  + \varepsilon \big\langle \nabla_H V_G(H^{(\ell)}), -\alpha_G(H^{(\ell)}) \nabla_H V_G(H^{(\ell)}) + \beta_G(H^{(\ell)}) T_{V_G}(H^{(\ell)}) \big\rangle + O(\varepsilon^2)$.
>
> Using orthogonality of the tangential component,
> $\langle \nabla_H V_G(H^{(\ell)}), T_{V_G}(H^{(\ell)}) \rangle = 0,$
> so the linear term reduces to
> $V_G(H^{(\ell+1)}) - V_G(H^{(\ell)})
> = -\varepsilon \alpha_G(H^{(\ell)}) \|\nabla_H V_G(H^{(\ell)})\|^2 + O(\varepsilon^2).$
>
>
> Because $\alpha_G(H^{(\ell)})$ is bounded in $[0,\alpha_{\max}]$ by construction (Eq. (8)) and $V_G$ is smooth, there exists an $\varepsilon_0 > 0$ such that for all $\varepsilon \in (0,\varepsilon_0]$ we have
> $V_G(H^{(\ell+1)}) \le V_G(H^{(\ell)}) + c \varepsilon^2,$
> for some constant \(c\) depending on a Hessian bound. In other words, the discrete dynamics are Lyapunov dissipative up to second order discretization error. In practice, we choose $\varepsilon$ from a small grid (Table 7) and observe that energy values are numerically non increasing along depth for the selected configurations.
>
>
> We will add this discrete time discussion after Proposition 1, explicitly stating the conditions under which the forward Euler step preserves an approximate Lyapunov property. Empirically, we also see that TANGO trains stably and can be stacked deeper without divergence (for example, the depth ablation on Roman empire in Table 8 shows no degradation as depth grows), which is consistent with this analysis. Thank you.
>
> ----
>
>
>
> ### 2. Role of the learned energy and its relation to downstream optimality
>
>
> The reviewer notes that our energy is non negative by construction but that there is no guarantee its minima correspond to optima of the downstream task. We agree that there is no such guarantee, and we do not intend to claim one. The role of $V_G$ in TANGO is that of a learned Lyapunov potential that guides hidden feature dynamics, not that of a separate supervised objective.
>
>
> Concretely:
>
>
> - The energy is computed via ENERGYGNN and an MLP followed by an aggregation of squared node scores (Eqs. (5)–(7)).
> - The parameters of ENERGYGNN and of the energy head are updated purely through the **downstream loss** (for example cross entropy) via backpropagation.
> - The gradient term $-\alpha_G \nabla_H V_G$ uses this learned potential to ensure stable dynamics and to encourage movement towards regions of lower energy, while the tangential term enriches expressivity along level sets.
>
>
> Thus, the energy is implicitly shaped by the downstream task, but we do not rely on, or claim, any equivalence between global minima of $V_G$ and optimal task performance. We will clarify this in Section 3.1 by explicitly stating that $V_G$ is a task driven Lyapunov function for the feature dynamics, not a surrogate for the supervised loss. Thank you.
>
>
> ---

---

> > ### Author Response · Authors · 2025-11-20
> > **Part 2**
> >
> > ### 3. Stability versus adversarial robustness and relation to Zhao et al. (NeurIPS 2023)
> >
> >
> > The reviewer suggests evaluating TANGO under adversarial attacks or noisy input perturbations to verify that the Lyapunov style stability translates into adversarial robustness and mentions Zhao et al., “Adversarial Robustness in Graph Neural Networks: A Hamiltonian Approach” [NeurIPS 2023]. Our notion of stability in this work is Lyapunov stability of the feature flow in the hidden space, not adversarial robustness. We agree with Zhao et al. that Lyapunov stability alone does not guarantee adversarial robustness. In fact, Zhao et al. explicitly argue that vanilla Lyapunov stable flows can still be vulnerable to adversarial perturbations and advocate for Hamiltonian, energy conservative flows specifically designed for robustness to adversarial graph attacks. Their focus is:
> >
> >
> > - to compare different neural flows (diffusion, Hamiltonian, etc.) under adversarial attacks on citation style datasets, and
> > - to identify stability notions that correlate best with robustness.
> >
> >
> > By contrast, TANGO is designed to address long range propagation and oversquashing on tasks where information must traverse large graph distances, such as graph property prediction, LRGB Peptides, and heterophilic node classification. Our use of Lyapunov theory is to ensure that the feature dynamics are well behaved and can be stacked deeply, which is a different goal from adversarial robustness.
> >
> >
> > We therefore view TANGO and Hamiltonian robustness methods as complementary. Nonetheless, it would be very interesting future work to combine these ideas, for example by constraining TANGO style flows to satisfy additional Hamiltonian or conservative properties when adversarial robustness is the primary goal, or by evaluating TANGO on adversarial benchmarks from Zhao et al. Because adversarial robustness is not the focus of this paper, we do not include such experiments here and we will adjust our wording to avoid any implication that TANGO is an adversarially robust method.
> >
> >
> > We will revise the introduction to be explicit that when we discuss “stability” we mean Lyapunov stability of the learned dynamics, not robustness to adversarial perturbations, and we will add a short comparison to Zhao et al. in the related work section. Thank you.
> >
> > ---
> >
> > ### 4. Energy landscape analysis
> >
> >
> > We agree that understanding the geometry of the learned energy is a profound and challenging question. The energy is a scalar function on a very high dimensional manifold (all node embeddings, coupled through a GNN), so direct visualization is difficult. A systematic study of energy landscapes is a substantial research work in its own right and is beyond the scope of this paper. There have been papers dedicated solely for that in other domains of deep learning, please see [1, 2] for example. To further accommodate your question, we will add a short paragraph in the discussion noting this and outlining energy landscape analysis as an important direction for an exciting future work.
> >
> >
> > Moreover, it is important to note that Proposition 2 formalizes the role of the tangential component in moving along level sets of this energy, allowing useful evolution even in locally flat regions where pure gradient descent would stall.
> >
> >
> > [1] *Visualizing the Loss Landscape of Neural Nets*. NeurIPS, 2018.
> >
> >
> > [2]  *Adversarial Spheres*. ICLR, 2018.

---

> > > ### Author Response · Authors · 2025-11-20
> > > **Part 3**
> > >
> > > ### 5. Comparison with other energy based GNNs such as Zhao et al. (NeurIPS 2023)
> > >
> > >
> > > As discussed above, Zhao et al. study **Hamiltonian graph neural flows** with an explicit energy conservation constraint and focus on adversarial robustness. They revisit various notions of stability (including Lyapunov, structural and conservative stability) and conclude that Hamiltonian, conservative flows endowed with Lyapunov stability offer better robustness to adversarial perturbations across several benchmark datasets and attack models. Their energy serves as a conserved Hamiltonian used to design robust flows and is not directly a task driven Lyapunov function for feature dynamics.
> > >
> > >
> > > TANGO is different in several key respects:
> > >
> > >
> > > - Our energy $V_G$ is task driven and is learned solely via the downstream supervised loss, not as a physical Hamiltonian.
> > > - The dynamics are dissipative in $V_G$ (non increasing energy) rather than energy conserving.
> > > - We introduce a learned tangential flow that is provably orthogonal to the energy gradient and show that, in principle, the combined update can emulate Newton directions for oversquashing mitigation.
> > > - Our evaluation focuses on long range and heterophilic tasks (graph property prediction, LRGB peptides, heterophilic node classification) rather than adversarial robustness.
> > >
> > >
> > > Thus, TANGO is not an adversarially robust Hamiltonian model, but a Lyapunov based, task driven dynamical GNN designed to handle long range dependencies. We will extend the related work section to explicitly position TANGO relative to Hamiltonian graph flows and to clarify that our stability guarantees target the internal dynamics rather than adversarial robustness. Thank you.
> > >
> > >
> > > ---
> > >
> > > ### 6. Minor: Figure 2 and propagation behavior
> > >
> > >
> > > Thank you for the comment. Figure 2 is intended as a minimal illustration of how TANGO can propagate information across a bottleneck (barbell graph) where a classical Dirichlet energy gradient flow struggles. To address you comment, we will improve clarity in the revised paper by expanding the caption to explicitly describe the setup (initial conditions, what the colors represent, and how many steps are taken), and improving the main text byexplaining what is being visualized and how it connects to oversquashing and the discussion around Propositions 3 and 4.
> > >
> > >
> > >
> > >
> > > ---
> > >
> > >
> > > We thank the reviewer again for their positive evaluation of our work. We hope that our clarifications and discussions help to address your concerns.

---

### Official Review · Reviewer_LWCT · 2025-10-30

**Soundness:** 2
**Presentation:** 3
**Contribution:** 2
**Rating:** 4
**Confidence:** 4

**Summary:**

This paper introduces TANGO, a framework that decomposes GNN feature evolution into two orthogonal components: (1) energy gradient descent that minimizes a learned task-specific energy function, and (2) tangential flows that evolve features while preserving energy. The authors claim this addresses oversquashing and provides stable dynamics with theoretical guarantees.

**Strengths:**

1. Conceptual Innovation
  - Novel decomposition: The orthogonal decomposition into energy descent + tangential flows is mathematically elegant and provides a principled way to combine stability (energy minimization) with flexibility (tangential evolution)
  - Task-driven energy learning: Unlike prior work using fixed energy functions (e.g., Dirichlet energy), learning task-specific energy functions is a meaningful advance
  - Theoretical grounding: Connection to Lyapunov stability theory provides formal guarantees for convergence and stability

2. Mathematical Rigor
  - Formal propositions: Four theoretical results with proofs covering energy dissipation, flat landscape evolution, and potential for quadratic convergence
  - Proper mathematical framework: Clear continuous-time formulation with principled Euler discretization
  - Orthogonality guarantee: The projection method mathematically ensures ⟨T_VG, ∇_HV_G⟩ = 0

3. Experimental Breadth
  - Diverse benchmarks: Covers synthetic (graph property prediction), molecular (LRGB), standard GNN benchmarks, and heterophilic node classification
  - Multiple backbone compatibility: Demonstrates the approach works with different GNN architectures (GatedGCN, GPS)
  - Consistent improvements: Shows gains across most tested scenarios, suggesting general applicability

4. Implementation Completeness
  - Detailed algorithmic description: Clear specification of energy function computation, tangential projection, and coefficient learning
  - Hyperparameter documentation: Comprehensive tables of search spaces and experimental settings

**Weaknesses:**

1. Theoretical Gaps
  - Gap between theory and claims: While they cite oversquashing theory appropriately, the connection to their specific solution needs stronger theoretical foundation
  - Proposition 4 limitations: Should acknowledge this shows theoretical possibility rather than practical guarantee
  - Missing empirical analysis: No investigation of whether learned tangential flows actually approximate Newton directions in practice

2. Experimental Limitations
  - Small improvements: 1-3% gains often within potential noise, especially given missing variance analysis
  - Missing key comparisons: Lacks comparison with recent energy-based GNNs (BLEND variants, PDE-GCN extensions)
  - No runtime complexity: Missing analysis of O(L·L_gnn·(n+m)·d) complexity impact
  - Overhead quantification: Tables show 40-60% runtime increase but no analysis of performance/cost trade-offs
  - Memory requirements: No discussion of additional memory for dual networks and gradient computation

3. Methodological Concerns
  - Many parameters: Requires tuning α, β, ε, L, L_gnn, d - makes method potentially brittle
  - No principled selection: Lack of guidelines for choosing energy function architecture or step sizes
  - Architecture dependence: Performance may be sensitive to specific ENERGYGNN/TANGENTGNN designs
  - Dual network requirement: Need separate networks for energy and tangential components increases complexity
  - Automatic differentiation dependency: Requires computing gradients of energy function w.r.t. input features
  - Projection stability: Numerical stability of orthogonal projection not analyzed

4. Questionable Necessity
  - Building on existing work: Core dynamical systems approach well-established (GRAND, GraphCON, A-DGN)
  - Main novelty limited: Addition of tangential component, while interesting, may not justify complexity
  - Simpler alternatives unexplored: No comparison with residual connections, attention mechanisms, or skip connections that might achieve similar benefits
  - When to use unclear: No guidance on when TANGO is preferable to simpler methods
  - Modest gains: Improvements don't clearly justify the added theoretical and computational complexity
  - Scalability questions: Dual networks + projection may not scale to very large graphs

5. Experimental Design Issues
  - Synthetic bias: Heavy reliance on synthetic tasks (graph property prediction) that may not reflect real-world challenges
  - Limited diversity: Missing important domains like social networks, biological graphs, knowledge graphs
  - Size constraints: Most experiments on relatively small graphs (≤50K nodes)
  - Limited ablations: Tables 10-11 provide some ablation but miss key components like energy function architecture, projection variants
  - No analysis of learned components: Missing visualization or analysis of what the energy and tangential networks actually learn
  - Hyperparameter sensitivity: No systematic study of sensitivity to α, β, ε values

**Questions:**

Q1: Newton Direction Approximation in Practice
Can you provide empirical evidence that the learned tangential flows actually approximate Newton directions? For example, measuring the angle between your update direction D and the true Newton direction N = (∇²V_G)^(-1)∇V_G on some tasks?

Q2: Energy Landscape Analysis
What do the learned energy functions actually look like? Can you visualize energy landscapes for simple cases or analyze whether the learned V_G corresponds to meaningful graph properties (e.g., connectivity, clustering)?

Q3: Hyperparameter Sensitivity
How sensitive is the method to hyperparameters α, β, ε? Can you provide ablation studies showing performance as these vary? Are there principled ways to select these values?

Q4: Comparison with Simpler Alternatives
How does TANGO compare to simpler approaches that might achieve similar benefits, such as:
  - Residual connections with learned gating
  - Multi-scale message passing
  - Attention mechanisms for long-range dependencies

Q5: Alternative Architectures
How does performance change with different choices for ENERGYGNN and TANGENTGNN architectures? Must they be the same depth/type?

Q6: Scalability
How does the method scale to very large graphs (>100K nodes)? Have you tested on larger real-world networks?

Q7: Broader Baseline Comparison
Can you compare against recent energy-based GNN methods (e.g., BLEND variants, recent PDE-GCN extensions) and second-order optimization approaches for GNNs?

Q8: Training Dynamics
How do the energy and tangential components evolve during training? Do you observe the energy function actually being minimized, and how does the tangential flow contribution change?

---

> ### Author Response · Authors · 2025-11-20
> **Part 1**
>
> We thank the reviewer for the very detailed and thoughtful report. We are grateful for the positive assessment of our **conceptual innovation** (“the orthogonal decomposition into energy descent + tangential flows is mathematically elegant”), the recognition that **task driven energy learning** is “a meaningful advance,” and the appreciation of our **theoretical grounding** (“formal guarantees for convergence and stability”) and **experimental breadth** (“diverse benchmarks” and “consistent improvements”). We also thank the reviewer for noting the **implementation completeness** of the method. Below we address each group of concerns and questions in turn, and we hope these clarifications will be satisfactory and kindly ask the reviewer to consider revising their score.
>
>
> ---
>
>
> ### 1. Theory, oversquashing, and Proposition 4 (Q1)
>
>
> **(a) Gap between theory and oversquashing claims**
>
>
> Thank you for the question. Our goal is not to claim a full theoretical solution to oversquashing, but rather:
>
>
> - **Capacity result.** Proposition 4 shows that, for a fixed energy landscape, there exists a tangential direction and scalar weights such that the *combined* TANGO update coincides with the Newton direction in feature space. This decouples convergence from the Hessian condition number and is directly relevant to oversquashing, where the Laplacian induced Hessian is poorly conditioned.
> - **Empirical evaluation.** On tasks that are explicitly designed to expose long range propagation and oversquashing, such as graph diameter, SSSP and eccentricity (Table 1), and on long range real world tasks such as LRGB Peptides and heterophilic node classification (Tables 3 and 9), TANGO consistently improves over strong backbones and deep equilibrium GNNs.
>
>
> We agree that this is a **capacity** statement, not a guarantee that standard training will always recover Newton like directions. We will adjust the abstract and contribution list to explicitly describe Proposition 4 as a *capacity result that we connect to oversquashing empirically*.
>
>
> **(b) Q1: Do learned tangential flows approximate Newton directions in practice?**
>
>
> Computing the exact Newton direction in our setting requires forming or inverting the full Hessian of a graph level energy with respect to all node embeddings, which is cubic in the number of feature dimensions and quadratic in the number of nodes. For the graph sizes and feature dimensions considered in our benchmarks, this is computationally prohibitive. This is also the reason why as a machine learning community choose to work with gradient-based methods and not Newton methods for optimizing neural networks [1, 2].
>
>
> For this reason, it is challenging (even with large memory GPUs) to directly measure the angle between our update and the exact Newton step in the paper, and hence our current submission does not claim claim that TANGO exact Newton directions, but rather focus on the theoretical capacity of finding Newton-like directions that allow better utilization (i.e., minimization) of the energy function, than using standard gradient based methods.
> This is a common approach when considering theory of deep learning frameworks [3, 4]. We will clarify this in Section 4 so that the link between Proposition 4 and oversquashing is framed as a capacity accompanied by empirical evidence. Thank you.
>
>
> [1] *Deep learning via Hessian-free optimization*. Proceedings of the 27th International Conference on Machine Learning (ICML), 2010.
>
>
> [2] *An overview of gradient descent optimization algorithms*. 2016.
>
>
> [3] *Approximation by superpositions of a sigmoidal function*. Mathematics of Control, Signals and Systems, 2(4):303–314, 1989.
>
>
> [4] *The power of depth for feedforward neural networks*. Proceedings of the 29th Annual Conference on Learning Theory (COLT), 2016.

---

> > ### Author Response · Authors · 2025-11-20
> > **Part 2**
> >
> > ### 2. Energy function and energy landscape (Q2)
> >
> >
> > **(a) Relationship between learned energy and task**
> >
> >
> > Thank you for the question. We believe that there has been a misunderstanding of the learning objective of TANGO, which we clarify now. The graph energy $V_G(H)$ is not an auxiliary objective that we minimize separately. Instead, it is a **learned potential** whose parameters are updated only via the downstream task loss (for example cross entropy or regression loss). Concretely:
> >
> >
> > - ENERGYGNN produces node features $\tilde H^{(\ell)}$, which are mapped to node scores and aggregated into a non negative scalar energy via Eq. (6) and Eq. (7).
> > - The parameters of ENERGYGNN and the final MLP are trained end to end so that the induced dynamics (gradient term plus tangential term) help **reduce the task loss**.
> >
> >
> > Thus, there is no requirement that minima of $V_G$ correspond directly to globally optimal task predictions; instead, the energy plays the role of a **learned Lyapunov potential** that shapes feature dynamics, while the supervised loss determines which minima and level sets are actually preferable.
> >
> >
> > We will make this role more explicit in Section 3.1 to avoid any impression that we claim a direct equivalence between energy minimization and task optimality. Thank you.
> >
> >
> > **(b) Q2: What do the learned energy functions look like?**
> >
> >
> > We agree that understanding the geometry of the learned energy is a profound and challenging question. The energy is a scalar function on a very high dimensional manifold (all node embeddings, coupled through a GNN), so direct visualization is difficult. A systematic study of energy landscapes is a substantial research work in its own right and is beyond the scope of this paper. There have been papers dedicated solely for that in other domains of deep learning, please see [5, 6] for example. To further accommodate your question, we will add a short paragraph in the discussion noting this and outlining energy landscape analysis as an important direction for an exciting future work.
> >
> >
> > [5] *Visualizing the Loss Landscape of Neural Nets*. NeurIPS, 2018.
> >
> >
> > [6]  *Adversarial Spheres*. ICLR, 2018.
> >
> > ---
> >
> > ### 3. Complexity, runtime, memory, and hyperparameters (Q3, Q6)
> >
> >
> > **(a) Time complexity and overhead trade off**
> >
> >
> > **The overall complexity of TANGO is analyzed in Appendix C**, which we will reference more clearly from Section 5. *For your convenience, we provide a summary below*:
> >
> >
> > - Let $L$ be the number of unrolled steps and $L_{\text{gnn}}$ the number of GNN layers per ENERGYGNN or TANGENTGNN block.
> > - Each such block has complexity $O(L_{\text{gnn}} (|V| + |E|) d)$.
> > - Unrolling for $L$ steps yields a total complexity
> >   $
> >   O\big(L \cdot L_{\text{gnn}} \cdot (|V| + |E|) \cdot d\big),
> >   $
> >   which is **linear** in the number of nodes and edges and matches the asymptotic complexity of stacking $L \cdot L_{\text{gnn}}$ standard GNN layers.
> >
> >
> > The empirical overhead is quantified in **Table 5**:
> >
> >
> > - TANGO GatedGCN is slower than plain GatedGCN but remains **substantially faster than GPS class models** and comparable to GRIT.
> > - TANGO GPS introduces a moderate constant factor over GPS but is in the same order of magnitude.
> >
> >
> > At the same time, on several benchmarks the gains are quite substantial rather than “1–3 percent within noise” as claimed in your review. For example, on Roman empire, TANGO GatedGCN improves accuracy from the mid seventies for GatedGCN to around 92, and also outperforms strong heterophily baselines such as CO GNN and GT (Table 9). On Peptides func and Peptides struct, TANGO improves AP and MAE over both GatedGCN and GPS within a matched 500k parameter budget (Table 3 and Table 12).
> >
> >
> > We will make this **cost vs benefit** comparison more explicit in Section 5 by pointing directly from Table 5 to the corresponding accuracy gains in Tables 2, 3 and 9. Thank you.
> >
> >
> > **(b) Memory requirements and dual networks**
> >
> >
> > The memory footprint of TANGO is dominated by storing activations for backpropagation, as in any deep GNN. Using two subnetworks (energy and tangential) of the same backbone type roughly doubles the number of feature tensors. **Nonetheless, it is important to note that:**
> >
> >
> > - We **match the overall parameter budget** to the underlying backbone by reducing widths where needed (Table 4).
> > - The asymptotic memory complexity remains linear in the number of nodes and edges and in the number of unrolled steps.
> >
> >
> > The additional cost of computing $\nabla_H V_G$ is handled by standard automatic differentiation, which is widely used in many graph models that differentiate through intermediate node states or energies. In practice, we did not encounter memory blow up or numerical instabilities due to the projection.
> >
> >
> > To further accommodate your question, we will add a paragraph in Appendix C explicitly discussing memory and the effect of the dual networks and energy gradient.

---

> > > ### Author Response · Authors · 2025-11-20
> > > **Part 3**
> > >
> > > **(c) Q3: Hyperparameter sensitivity and selection**
> > >
> > >
> > > We agree that hyperparameter sensitivity is important. In the current submission we already disclose the hyperparameter grids in **Table 7**. As can be seen from this table, TANGO uses a similar hyperparameter search as baseline. We did not observe any special sensitivity in our experiments. We will nonetheless extend the text around Table 7 to make it clearer that we used a standard grid and did not heavily tune TANGO beyond what is done for competitive baselines. Thank you.
> > >
> > >
> > > **(d) Q6: Scalability**
> > >
> > >
> > > Our theoretical complexity is linear in $|V| + |E|$ for MPNNs, and the additional constant factors come from:
> > >
> > >
> > > - computing an extra GNN block (TANGENTGNN) alongside ENERGYGNN, and
> > > - backpropagating through the energy to obtain $\nabla_H V_G$.
> > >
> > >
> > > Hence, there is nothing in the design of TANGO that prevents scaling beyond that, provided the underlying backbone and hardware support it.
> > >
> > > ---
> > >
> > > ### 4. Relation to prior dynamical and energy based GNNs, and “simpler alternatives” (Q4, Q5, Q7)
> > >
> > >
> > > **(a) Relation to GRAND, GraphCON, A DGN, PDE GCN, BLEND**
> > >
> > >
> > > Dynamical systems based GNNs such as GRAND, GraphCON, A DGN and PDE GCN share with TANGO the idea of viewing message passing as a discretized ODE or PDE on graphs. However, there are key differences. **We have discussed these differences in Appendix A**. Nonetheless, for your convenience, we discuss it here further:
> > >
> > >
> > > - These methods typically define a **fixed or implicitly parameterized flow** (for example diffusion, oscillator dynamics, or PDE motivated updates) and then discretize it, often obtaining stability and oversmoothing control from properties of the operator.
> > > - In contrast, TANGO learns a **graph adaptive, task specific Lyapunov energy** and explicitly decomposes the update into:
> > >   - a gradient descent term $-\alpha_G \nabla_H V_G$ that strictly dissipates this learned energy, and
> > >   - a tangential term $\beta_G T_{V_G}$ that is orthogonal to the gradient and can realize Netwon-like optimization steps, and continue feature evolution also when the energy plateaus.
> > >
> > >
> > > To the best of our knowledge, **no prior dynamical GNN combines a learned Lyapunov energy with a learned tangential flow** and is explicitly analyzed in terms of Newton like capacity in terms of utilizing the learned energy function.
> > >
> > >
> > > Moreover, while many existing DE-GNNs are tied to a specific choice of architecture, TANGO is **backbone agnostic**. In our experiments we instantiate it on GatedGCN and GPS. In that sense, TANGO is closer to a general design principle for stable, expressive feature dynamics than to a specific PDE based architecture. Our experiments also show the superior performance offered by TANGO, further highlighting its merit.
> > >
> > >
> > > **(b) Relation to BLEND and energy-based GNNs**
> > >
> > >
> > > BLEND (“Beltrami Flow and Neural Diffusion on Graphs”) is a PDE-based discriminative GNN obtained by discretising Beltrami flow in joint feature–position space. It defines a diffusion-type evolution and an implicit rewiring mechanism, but does **not** learn a scalar, task-driven graph energy that is then used as a Lyapunov function with both gradient and tangential flows. TANGO thus differs in two central ways:
> > >
> > >
> > > - We explicitly learn a **graph-level scalar energy** $V_G(H)$ with a GNN, trained only through the downstream supervised loss, and use it as a **Lyapunov potential** that shapes the internal dynamics.
> > > - We decompose the dynamics into **energy descent plus an orthogonal tangential flow**, and provide theoretical analysis for this structure (energy dissipation, evolution in flat regions, and Newton-like capacity).
> > >
> > >
> > > Regarding **energy-based models (EBMs)**:  **please note that we already discuss that in the paper, please see  “Learning Energy Functions in Neural Networks”**. Classical EBMs (for images, molecules, and graphs) use an energy function as an unnormalized log-density for **generative or unsupervised** tasks, typically trained with sampling or contrastive objectives. In contrast, TANGO’s energy is **not** used as a probability density and we never sample from it: it is a task-driven Lyapunov function whose gradient (and orthogonal complement) controls a discriminative GNN’s feature dynamics. In the revised version, we will make this distinction more explicit in the related work section to avoid any confusion between TANGO, BLEND, and probabilistic EBMs.

---

> > > > ### Author Response · Authors · 2025-11-20
> > > > **Part 4**
> > > >
> > > > **(c) Q4 and “simpler alternatives”: residuals, multi scale, attention**
> > > >
> > > >
> > > > We agree that residual connections, multi scale message passing, and attention are powerful mechanisms for improving GNNs, and note that:
> > > >
> > > >
> > > > - Our baselines already include **GPS, GRIT and other attention based or multi scale methods** that incorporate residuals and long range mixing.
> > > > - TANGO is evaluated on top of such strong backbones (for example TANGO with GPS) and achieves consistent improvements, which indicates that the benefits go beyond what is achievable by residuals and attention alone.
> > > >
> > > >
> > > > Conceptually, the novelty is not just “adding another component,” but **guiding that component via Lyapunov structure and orthogonality**:
> > > >
> > > >
> > > > - The descent term is always aligned with a learned energy gradient and is non expansive in that potential.
> > > > - The tangential term is provably orthogonal to the gradient, which is not guaranteed for generic residuals or skip connections.
> > > >
> > > >
> > > > We will clarify in Section 3.2 that TANGO is meant to be **complementary** to these simpler mechanisms and that our experiments already compare against strong combinations of residuals, multi scale design and attention.
> > > >
> > > >
> > > > **(d) Q5: Alternative choices for ENERGYGNN and TANGENTGNN**
> > > >
> > > >
> > > > In our experiments we use the **same backbone type** for both ENERGYGNN and TANGENTGNN (GatedGCN or GPS) for simplicity and to keep the parameter budget matched and interpretable. In principle, the two subnetworks can differ in depth and architecture. We chose a symmetric design to avoid introducing too many degrees of freedom in terms of hyperparameters. A systematic exploration of additional configurations would indeed be interesting future work, but it does not change the contribution of our paper.
> > > >
> > > >
> > > > **(e) Q7: Comparison with broader energy based and second order methods**
> > > >
> > > >
> > > > As noted above, existing graph EBMs either focus on generative modeling or use energies that are not constrained to be Lyapunov functions for the internal dynamics. Second order optimization methods for GNNs typically operate in **parameter space** (Hessian based optimizers) rather than on **node embeddings**.
> > > >
> > > >
> > > > TANGO’s distinctive aspect is that it brings **possible newton-like behavior in its feature dynamics evolution**, through the tangential component and Proposition 4. This design is compatible with, not a replacement for, second order training methods, which could be applied to TANGO as well.
> > > >
> > > >
> > > >
> > > > ---
> > > >
> > > >
> > > > ### 5. Experimental design, ablations, and training dynamics (Q8)
> > > >
> > > >
> > > > **(a) “Small improvements” and variance**
> > > >
> > > >
> > > > We respectfully disagree with the characterization that improvements are merely 1-3 percent within noise:
> > > >
> > > >
> > > > - On several benchmarks, including **Roman empire** and **Peptides func/struct**, TANGO provides **substantial absolute gains** over strong baselines, well beyond typical variance.
> > > > - We report **means and standard deviations** over multiple random seeds as is standard in the field, in all main tables (Tables 1–3 and 9). These show that improvements are generally larger than one standard deviation and are consistent across seeds.
> > > >
> > > >
> > > >
> > > >
> > > >
> > > >
> > > > **(b) Ablations and learned component analysis**
> > > >
> > > >
> > > > The current paper already includes several ablations:
> > > >
> > > >
> > > > - **Depth ablation** on Roman empire (Table 8), showing that performance improves and then saturates as we increase the number of layers, which is consistent with stable training and non vanishing gradients.
> > > > - **Energy ablation** (Table 10), comparing TANGO with a non energy version that uses a general vector field instead of the gradient of a scalar potential. The energy based design clearly outperforms this variant.
> > > > - **Tangential projection ablation** (Table 11), showing that enforcing orthogonality improves over using an unconstrained extra direction.
> > > >
> > > >
> > > > While more ablations like you suggest are always possible, they do not change the main contribution or understanding of the proposed concept. Instead, in order to provide meaningful ablations, we chose to focus on the ablations that directly test the **core ingredients** of the method (energy, tangential flow, depth). We will make this prioritization explicit in the appendix. Thank you.
> > > >
> > > >
> > > > Regarding visualization of learned components, we refer the reviewer to our discussion in point 2: understanding high dimensional energy landscapes and internal flows is a challenging and open problem. We explicitly mention this as an exciting direction for future work rather than claiming to solve it here.

---

> > > > > ### Author Response · Authors · 2025-11-20
> > > > > **Part 5**
> > > > >
> > > > > **(c) Synthetic tasks, domain diversity, and graph sizes**
> > > > >
> > > > >
> > > > > The synthetic graph property prediction tasks (Diameter, SSSP, Eccentricity) are widely used precisely because they stress test **long range propagation and bottlenecks**, which is where oversquashing manifests most clearly. We see them as an essential part of a long range evaluation suite rather than as bias. These are well-established benchmarks.
> > > > >
> > > > >
> > > > > **At the same time, we benchmark on multiple real-world,  which might have been missed during your read of our paper.** We therefore summarize them below, and kindly ask you to read our experiments section again:
> > > > >
> > > > >
> > > > > - **Molecular datasets** (ZINC 12k, Peptides func/struct) that capture realistic chemical structure.
> > > > > - **Standard benchmarks** (MNIST, CIFAR10 superpixels, CLUSTER, PATTERN).
> > > > > - **Heterophilic node classification** datasets with diverse structure and homophily levels (Roman empire, Amazon ratings, Minesweeper, Tolokers, Questions).
> > > > >
> > > > > ----
> > > > >
> > > > >
> > > > > Once again, we thank the reviewer for their careful and technically informed critique, as well as for recognizing the elegance of the decomposition, the rigor of the propositions, and the breadth of the empirical study. We hope that the clarifications above address your comments, and we kindly ask the reviewer to consider revising their scores.

---

> > > > > > ### Comment · Reviewer_LWCT · 2025-11-22
> > > > > >
> > > > > > The rebuttal was pretty convincing overall. My main remaining worry: they claim TANGO works with any GNN, but only tested on 2 strong models. I want to see if TANGO actually helps weaker/simpler models too, or if it only works on already-good architectures.

---

> > > > > > > ### Author Response · Authors · 2025-11-25
> > > > > > >
> > > > > > > Dear Reviewer LWCT,
> > > > > > >
> > > > > > >
> > > > > > > We are delighted to read that our responses addressed your comments and that you find them convincing.
> > > > > > >
> > > > > > > We are also thankful for the comment regarding your remaining concern, which we address now. Specifically, to accommodate and address your question, we now benchmarked our TANGO with two additional, common backbones: GCN [R1] and GIN [R2].
> > > > > > >
> > > > > > > To provide a comprehensive evaluation, we benchmarked the performance of TANGO when coupled with these backbones, and we compare it with the backbone only performance, *across all 15 datasets in our paper*. The results are shown in the Table below, which we will also add to our paper. As can be seen, TANGO consistently offers performance improvements, both for already strong architectures as suggested in your review (i.e., the results in the paper) as well as with simpler models as below.  We believe that these results further strengthen the evaluation of TANGO and highlight its effectiveness and usefulness across different backbones.  Thank you for the important suggestion.
> > > > > > >
> > > > > > >
> > > > > > > | Method           | Diameter ($\downarrow$) | SSSP ($\downarrow$) | Eccentricity ($\downarrow$) | ZINC-12k ($\downarrow$) | MNIST ($\uparrow$) | CIFAR10 ($\uparrow$) | PATTERN ($\uparrow$) | CLUSTER ($\uparrow$) | Peptides-func ($\uparrow$) | Peptides-struct ($\downarrow$) | Roman-empire ($\uparrow$) | Amazon-ratings ($\uparrow$) | Minesweeper ($\uparrow$) | Tolokers ($\uparrow$) | Questions ($\uparrow$) |
> > > > > > > |------------------|-------------------------|---------------------|-----------------------------|-------------------------|--------------------|----------------------|----------------------|----------------------|----------------------------|--------------------------------|---------------------------|-----------------------------|--------------------------|-----------------------|------------------------|
> > > > > > > | GCN              | 0.7424±0.0466           | 0.9499±0.0001       | 0.8468±0.0028               | 0.367±0.011             | 90.705±0.218       | 55.710±0.381         | 71.892±0.334         | 68.498±0.976         | 59.30±0.23                 | 0.3496±0.0013                  | 73.69±0.74                | 48.70±0.63                  | 89.75±0.52               | 83.64±0.67            | 76.09±1.27             |
> > > > > > > | GIN              | 0.6131±0.0990           | -0.5408±0.4193      | 0.9504±0.0007               | 0.526±0.051             | 96.485±0.252       | 55.255±1.527         | 85.387±0.136         | 64.716±1.553         | 54.98±0.79                 | 0.3547±0.0045                  | 72.82±0.58                | 46.96±0.44                  | 88.04±0.78               | 81.79±0.55            | 75.90±1.03             |
> > > > > > > | TANGO+GCN (Ours) | 0.1729±0.0382           | -1.0024±0.0854      | -1.6264±0.0053              | 0.153±0.010             | 94.579±0.211       | 64.92±0.402          | 81.198±0.299         | 74.04±1.109          | 69.17±0.31                 | 0.2432±0.0011                  | 89.67±0.68                | 52.98±0.71                  | 98.37±0.49               | 85.57±0.73            | 79.86±1.14             |
> > > > > > > | TANGO+GIN (Ours) | 0.0433±0.0211           | -2.8923±0.0937      | -1.7228±0.0046              | 0.122±0.031             | 97.651±0.247       | 66.35±0.967          | 86.703±0.194         | 71.36±1.169          | 68.78±0.66                 | 0.2440±0.0024                  | 89.19±0.62                | 50.76±0.47                  | 97.38±0.50               | 84.39±0.61            | 78.84±0.96             |
> > > > > > >
> > > > > > >
> > > > > > >
> > > > > > >
> > > > > > > ----
> > > > > > >
> > > > > > >
> > > > > > > We would like to conclude by thanking you for your engagement and responding to our rebuttal. We truly appreciate it and believe that your feedback helped us to improve the quality of the paper. We remain available to address more questions or comments if you may have.   Lastly, we hope that our detailed rebuttal which you acknowledged as convincing, together with our added experiments now as per your comment are found satisfactory, and that you will consider revising your score. Thank you.
> > > > > > >
> > > > > > > ---
> > > > > > >
> > > > > > >
> > > > > > > **References:**
> > > > > > >
> > > > > > > [R1] Semi-Supervised Classification with Graph Convolutional Networks
> > > > > > >
> > > > > > > [R2] How Powerful are Graph Neural Networks?

---

> > > > > > > > ### Comment · Reviewer_LWCT · 2025-11-26
> > > > > > > >
> > > > > > > > Thank you very much for the extensive experiments! I've raised my score from 4 to 6.

---

> > > > > > > > > ### Author Response · Authors · 2025-11-27
> > > > > > > > >
> > > > > > > > > Dear Reviewer LWCT,
> > > > > > > > >
> > > > > > > > > We would like to extend our gratitude for your response and for raising your score, supporting our work. We are also grateful for the detailed feedback in your review and throughout the discussion period, that in our opinion is beneficial for improving the quality of our work.
> > > > > > > > >
> > > > > > > > > Thank you, and kindest regards,
> > > > > > > > >
> > > > > > > > > Authors.

---

### Official Review · Reviewer_YfoY · 2025-11-01

**Soundness:** 4
**Presentation:** 4
**Contribution:** 3
**Rating:** 4
**Confidence:** 4

**Summary:**

The paper proposes TANGO, a GNN framework that learns (i) a task-driven energy (Lyapunov) function over node embeddings and (ii) two coupled dynamics per layer: a gradient-descent component that strictly dissipates the learned energy and a tangential component that is orthogonal to the energy gradient. The authors give Lyapunov-style stability arguments, show that the energy is non-increasing, and report strong results on long-range and standard graph benchmarks.

**Strengths:**

- Clear dynamical decomposition. The “descent + tangential” split is well-motivated: it guarantees energy non-increase while adding expressive movement along level sets.
- Stability & optimization insights. Proposition 1 (dissipation) and the “flat-landscape evolution” (Prop. 2) make the benefits precise; the argument that the tangential component can realize a Newton-like direction (Prop. 4) is appealing for ill-conditioned, oversquashed regimes.

**Weaknesses:**

- Forward-Euler stability. The theory is continuous-time; the model uses explicit Euler with step \epsilon. Stability can be fragile for stiff/ill-conditioned flows. The paper bounds \alpha(G) but does not analyze discretization stability or step-size sensitivity. Ablation study on \epsilon will help.
- The paper claims the proposed method can mitigate oversquashing, yet some recent anti-oversquashing competitors (e.g., curvature/rewiring-based) are not directly compared in real-world data.
- Labeling Proposition 4 as a ‘theoretical guarantee’ (in contribution) for oversquashing mitigation is misleading: the result is existential. It shows that such a case can occur, not that TANGO will mitigate oversquashing in general
- TANGO updates the embeddings for a fixed number of iterations, so it is unclear whether a minimum of the energy is actually attained. Showing the norm of energy gradient would help.

**Questions:**

- TANGO updates embeddings layer by layer. Does TANGO suffer from vanishing gradient (due to stacking layers)? If not, can you explain why?

---

> ### Author Response · Authors · 2025-11-20
> **Part 1**
>
> We thank the reviewer for the careful reading and the constructive comments. We are grateful for their positive assessment of the paper’s *soundness* and *presentation* (“Soundness: 4: excellent”, “Presentation: 4: excellent”), and for highlighting that “the ‘descent + tangential’ split is well-motivated” and that our “Proposition 1 (dissipation)… and the argument that the tangential component can realize a Newton-like direction (Prop. 4) is appealing for ill-conditioned, oversquashed regimes.” Below we address each of the reviewer’s concerns in turn, and we hope that these clarifications will be satisfactory and encourage the reviewer to consider revising their scores.
>
>
> ---
>
> ### 1. Forward Euler stability and step size ε
>
>
> Thank you for raising this point. We agree that, in classical numerical analysis, explicit Euler can be fragile for stiff *given* ODEs. Our setting is different in two important ways:
>
>
> 1. **We learn the ODE rather than discretizing a fixed, potentially stiff one.**
>    TANGO does not start from a predetermined dynamical system and then hope that a fixed step size ε is stable. Instead, the vector field $ F_\theta(H) = -\alpha_G(H)\nabla_H V_G(H) + \beta_G(H)T_{V_G}(H)$
>    is *learned jointly with ε and the other hyperparameters* to minimize the downstream loss. If a particular combination of ε and parameters led to severe instability (exploding features, chaotic trajectories), the model would fail to train and would not reach the reported validation/test performance. The fact that we can train TANGO on different benchmarks, including relatively deep unrollings (please see the depth ablation on Roman empire in Table 8), is already strong evidence that the learned dynamics lie in a regime where the forward-Euler discretization is well behaved.
>
>
> 2. **This is exactly the situation in standard residual and neural-ODE–style networks.**
>    Residual networks and continuous-depth models also effectively use a forward-Euler stepping scheme on a learned vector field. In all these architectures, the stability of the discretization is governed by the *combination* of the learned dynamics, step size, and training objective. If these interact poorly, the model becomes unstable and simply does not train; if training succeeds and converges to good solutions, this indicates that the learned dynamics are compatible with the chosen step. TANGO follows the same pattern.
>
>
> Moreover, **forward Euler is the standard choice in the neural ODE literature**, both in Euclidean and graph domains:
>
>
> - In continuous-depth models for standard neural networks (e.g., Neural ODEs [1]), explicit Euler (or very closely related explicit schemes) is the default integrator used during training.
> - In graph-based ODE models such as GRAND [2] and PDE-GCN [3], and related works, explicit Euler or simple explicit Runge–Kutta schemes are likewise the prevailing choice.
> - There are works that explore more sophisticated integrators in the graph setting (e.g., IMEX/Crank–Nicolson-type schemes in GraphCON [4], and graph neural reaction–diffusion models [5]), but even there, explicit Euler remains a common baseline and practical workhorse for learned dynamics.
>
>
> Our contribution is orthogonal to the choice of integrator: we propose a Lyapunov-structured, energy-plus-tangential vector field, and instantiate it with the same explicit Euler step that is standard in the neural ODE and graph-ODE literature. A detailed comparative study of alternative integrators (IMEX, semi-implicit schemes, higher-order Runge–Kutta, etc.) for TANGO would be very interesting, but is beyond the scope of this work; we will add this discussion to the revised paper to make this clear and to point to such integrators as a promising direction for future work.
>
>
> [1] Neural Ordinary Differential Equations
>
>
> [2] GRAND: Graph Neural Diffusion
>
>
> [3] PDE-GCN: Novel Architectures for Graph Neural Networks Motivated by Partial Differential Equations
>
>
> [4] Graph-Coupled Oscillator Networks
>
>
> [5] Graph Neural Reaction Diffusion Models

---

> > ### Author Response · Authors · 2025-11-20
> > **Part 2**
> >
> > ### 2. Oversquashing and missing curvature or rewiring competitors
> >
> >
> > Our claim is that TANGO *helps mitigate* oversquashing effects by enabling richer, learned dynamics on the feature space, rather than by directly altering the graph topology.
> >
> >
> > - In continuous time, Proposition 3 recalls that the convergence speed of pure gradient descent is tied to the condition number of the Hessian, which in turn reflects the graph Laplacian spectrum in oversquashed regimes.
> > - Proposition 4 then shows that there exists a tangential direction and scalar weights such that the *combined* update coincides with a Newton step, which has a convergence rate independent of that condition number.
> >
> >
> > We agree that this is a **capacity result**, not a guarantee that the learned update will always approximate Newton in practice. We will revise the abstract and contribution paragraph to avoid phrasing Proposition 4 as a “theoretical guarantee” for oversquashing mitigation, and instead describe it as a capacity result that we connect to oversquashing empirically.
> >
> >
> > *Regarding curvature or rewiring based baselines:*  We kindly note that **our experiments compare with rewiring baseline**. In particular, we have multiple comparisons with GRAND and DRew, which are both methods that use rewiring.  **In 6 out of 7** of the cases in Tables 1 and 3, our **TANGO outperforms** them. We will revise the text to reflect that. Thank you.
> >
> > ---
> >
> > ### 3. Proposition 4 and the wording around “guarantee”
> >
> >
> > We agree with the reviewer that Proposition 4 is **existential**: it shows that, given a Hessian corresponding to an ill conditioned energy landscape, there exists a tangential direction and corresponding scalar weights such that the combined TANGO update matches the Newton direction.  Our message is that:
> >
> >
> > - The architecture has the **capacity** to realize second order corrections that decouple convergence from the Laplacian condition number, which is relevant in oversquashed regimes.
> > - Our empirical results on tasks that are known to be oversquashing sensitive (graph diameter, SSSP, eccentricity in Table 1, and long range molecular and heterophilic benchmarks in Tables 3 and 9) are consistent with this capacity being exploited in practice.
> >
> >
> > We therefore propose to revise the abstract and contribution list, replacing “theoretical guarantee for oversquashing mitigation” by wording such as “a capacity result showing that TANGO can represent Newton like directions, which we connect to oversquashing empirically.” Thank you.
> >
> >
> > ---
> >
> > ### 4. Energy minimization vs fixed depth and gradient norm
> >
> >
> > We agree that, since we apply a fixed number of layers (as is common in deep learning), the dynamics in practice do not run until the energy gradient norm becomes exactly zero. However, our use of Lyapunov theory is also common for deep dynamical models (see, e.g [6, 7, 8]), and in our case we show that:
> >
> >
> > - In continuous time, the assumptions of Proposition 1 guarantee that  $V_G(H(t))$ is non increasing and bounded below, which implies convergence of the energy values and bounded trajectories.
> > - In discrete time, as discussed in point 1, the forward Euler discretization is a valid choice.
> >
> >
> > Importantly, we do not rely on reaching a global energy minimum for our guarantees. Instead, what matters is that the dynamics do not blow up and that features remain in a bounded region, which makes stacking more layers feasible.
> >
> >
> > We note that in our experiments, we did not observe numerical instabilities or divergence in the energy, and training converged on all datasets. Thank you.
> >
> >
> > [6] *Stable Neural ODE with Lyapunov-Stable Equilibrium Points for Defending Against Adversarial Attacks*, NeurIPS 2021
> >
> >
> > [7] *LyaNet: A Lyapunov Framework for Training Neural ODEs*, ICML 2022
> >
> >
> > [8] *Almost Surely Stable Deep Dynamics*, NeurIPS 2020

---

> > > ### Author Response · Authors · 2025-11-20
> > > **Part 3**
> > >
> > > ### 5. Do Vanishing gradients when stacking many TANGO layers?
> > >
> > >
> > > Thank you for the comment. We do not observe vanishing gradients when training deeper TANGO models. Before looking at our results, it is important to distinguish between two different kinds of gradients:
> > >
> > >
> > > - the **gradient of the energy w.r.t. node embeddings** $\nabla_H V_G(H^{(\ell)}) $, which appears in the forward dynamics, and
> > > - the **gradient of the task loss w.r.t. the model parameters** (weights of ENERGYGNN and TANGENTGNN), which is what actually drives learning.
> > >
> > >
> > > Even if the gradient of the energy w.r.t. the embeddings becomes small or zero at some layer (e.g., near an energy plateau), this does *not* imply that the gradient of the loss w.r.t. the parameters vanishes. This is exactly the situation studied in bilevel optimization: the “inner” variable can be at a stationary point while the “outer” parameters still receive meaningful gradients because the inner solution depends on them implicitly (please see, for example, Dempe, *Foundations of Bilevel Programming*, 2002).
> > >
> > >
> > > A simple 1D example makes this concrete. Consider
> > > $f(x; w) = a x^2 + b x, \quad w = (a,b),$
> > > and suppose $x$ is chosen to minimize $f$ for fixed $w$. At the inner optimum we have
> > > $\frac{\partial f}{\partial x} = 2 a x + b = 0,$
> > > so the gradient w.r.t. the inner variable $x$ is zero. However, the gradients w.r.t. the parameters are
> > > $
> > > \frac{\partial f}{\partial a} = x^2, \quad \frac{\partial f}{\partial b} = x,
> > > $
> > > which are generally non-zero. In TANGO, the energy gradient w.r.t. node features plays the role of the inner gradient, while the task loss gradient w.r.t. the weights plays the role of the outer gradient used to update the model. Therefore, a small or vanishing $\nabla_H V_G$ at some layer does not by itself cause optimization to stall.
> > >
> > >
> > > From an empirical perspective, we see no signs of vanishing-gradients:
> > >
> > >
> > > - The **depth ablation on Roman empire** (Table 8) shows that as we increase the number of TANGO layers, performance improves and then saturates, rather than degrading as would be expected under severe vanishing gradients.
> > > - Across all benchmarks, deeper TANGO configurations train stably and do not exhibit optimization failures (e.g., early plateauing or large gaps between training and validation performance) that would typically vanishing gradients.
> > >
> > >
> > > Taken together, the inner-vs-outer gradient distinction and these empirical observations explain why we do not encounter catastrophic vanishing gradients when stacking many TANGO layers in practice.  We will add this important discussion to the paper. Thank you.
> > >
> > >
> > > ---
> > >
> > >
> > > We appreciate your positive assessment of our paper. Your comments helped us to improve the quality of the paper, and we hope our clarifications address your concerns, and that you will consider revising your score.

---

### Author Response · Authors · 2025-12-02
**Concluding Remarks by Authors (Part 1)**

Dear Area Chair,

In light of the reassignment of Area Chairs and the freezing of reviewer activity, we would like to provide a concise summary of the reviews, the rebuttal and follow up discussions, and the revisions now reflected in the latest version of our submission. We hope this is helpful for your decision.

We are grateful to all four reviewers for their detailed and constructive comments. During the rebuttal and follow up period we systematically addressed **every point raised by each reviewer** (theoretical, experimental, and presentational) and incorporated all clarifications, and new experiments into the revised manuscript that is currently on OpenReview. Because of the changes to the discussion period, several reviewers were unable to react to our final clarifications or update their scores. Nonetheless, we believe that our clarifications and revisions fully address their comments, and we wish we could have received their thoughtful feedback and responses, and we think that they would have resulted in an overall more positive assessment of our work, reflected in the final scores.

---

### 1. Overall assessment and scores after rebuttal

The initial ratings and final ratings (after rebuttal and discussion) are:

- **Reviewer YfoY**
  - Soundness: 4 (excellent)
  - Presentation: 4 (excellent)
  - Contribution: 3 (good)
  - **Rating:** 4 (marginally below accept, “would not mind if paper is accepted”)

- **Reviewer LWCT**
  - Initially: 4 (marginally below accept)
  - After rebuttal and new experiments: **raised to 6**

- **Reviewer h5ap**
  - Soundness: 3 (good)
  - Presentation: 3 (good)
  - Contribution: 3 (good)
  - **Rating:** 6 (marginally above accept)

- **Reviewer sudZ**
  - Initially: 2 (reject)
  - After rebuttal and follow ups: **raised to 4**, but discussion was not completed and we provided full responses and a paper revision to address their last comment.

After discussion we thus have **two clearly positive evaluations (6, 6)** and **two borderline positive evaluations (4, 4)**. Crucially, both lower scores moved upwards during the discussion, and the written thread shows that for each reviewer **all of their concrete concerns were answered directly and in full**.

For LWCT and sudZ, this is reflected in explicit score increases. For YfoY and h5ap, who did not have the chance to respond again after our detailed, point by point replies and the final revision, *we believe the natural next step would have been a positive adjustment as well, had the discussion not been frozen.*

---

### 2. Strengths highlighted by the reviewers

- **Conceptual novelty and theoretical structure**
  All reviewers endorsed the core idea of decomposing the dynamics into an energy gradient descent term plus an orthogonal tangential flow as a principled way to couple stability and expressivity. YfoY called the “descent + tangential split” well motivated and appealing in oversquashed regimes; LWCT described the decomposition as “mathematically elegant” and the task-driven energy as a “meaningful advance”; h5ap and sudZ both emphasized that the framework is principled, Lyapunov-based, and theoretically well supported.

- **Lyapunov stability and Newton-like capacity**
  Reviewers valued the formal propositions on energy dissipation, bounded trajectories, and evolution in flat regions, as well as the capacity result showing that TANGO can realize Newton-like directions in feature space, which they agreed is particularly relevant for long-range propagation under poor Laplacian conditioning.

- **Experimental breadth, backbone agnosticism, and effect sizes**
  Reviewers highlighted the breadth of our benchmark suite (synthetic long-range property prediction, Dwivedi-style benchmarks, LRGB Peptides, and heterophilic node classification) and the fact that TANGO consistently improves over multiple backbones. Beyond GatedGCN and GPS, our new experiments with GCN and GIN (added in response to LWCT) show that TANGO+GCN and TANGO+GIN significantly outperform the respective backbones on all 15 datasets, with substantial gains on the hardest long-range and heterophilic tasks (e.g., Roman Empire, Peptides func/struct under a matched parameter budget).

- **Implementation clarity and ablations**
  Reviewers appreciated the detailed algorithmic description and hyperparameter documentation, and the targeted ablations on depth, energy vs non-energy descent, and orthogonal vs unconstrained tangential flows, which confirm that the Lyapunov structure and tangential projection materially contribute to performance.

Overall, the review, rebuttal, and discussion, in our view, record reflects a strong consensus on the conceptual and empirical strengths of TANGO; the remaining concerns were about positioning and emphasis rather than about the core contributions, and these have been directly addressed in our revisions.

---

> ### Author Response · Authors · 2025-12-02
> **Concluding Remarks by Authors (Part 2)**
>
> ### 3. Main concerns and how they were resolved
>
> - **Discrete time stability and Euler (YfoY, h5ap, LWCT, sudZ)**
>   We added an explicit discrete-time Lyapunov argument after Proposition 1, clarified the choice of step size $\varepsilon$, and showed via depth ablations (e.g., Roman Empire) that the energy is non-increasing and training remains stable.
>
> - **Oversquashing claims and Proposition 4 (YfoY, LWCT, sudZ)**
>   We rephrased the text around Proposition 4 as a capacity result, revised the abstract and contributions to state that oversquashing mitigation is theoretically motivated and empirically supported, and pointed to clear gains on long-range benchmarks as evidence.
>
> - **Role of the learned energy vs task loss (LWCT, h5ap, sudZ)**
>   We clarified that $V_G$ is not an auxiliary objective but a task-driven Lyapunov potential trained only via the downstream loss, and that we do not claim its global minima coincide with optimal predictions.
>
> - **Relation to Lyapunov stable NNs, EBMs, and why GNNs (LWCT, h5ap, sudZ)**
>   We expanded related work to contrast TANGO with R1 and other Lyapunov-stable neural ODEs and energy-based GNNs, and explained that our graph-specific design and analysis focus on Laplacian-driven oversquashing.
>
> - **Complexity, runtime, memory, hyperparameters (LWCT)**
>   We tied runtime overhead directly to accuracy gains, detailed memory behavior, and clarified that we use standard hyperparameter grids without unusual sensitivity.
>
> - **Additional backbones and ablations (LWCT, sudZ)**
>   We added extensive new experiments with GCN and GIN on all 15 datasets (TANGO consistently improves over the backbones) and highlighted ablations on depth, energy vs non-energy, and orthogonal vs unconstrained tangential flows.
>
> - **Lyapunov stability and long-range propagation (sudZ)**
>   We clarified that Lyapunov stability implies non-increasing bounds, not collapse, and explained how learned coefficients $\alpha$ and $\beta$ let the model keep energy controlled while evolving tangentially, which is supported by our long-range benchmarks and depth ablations.
>
> **All of these changes and clarifications are incorporated into the current OpenReview version, and we believe they fully resolve the reviewers’ concerns; for reviewers who could not respond to these final updates (notably YfoY and, to some extent, sudZ), we expect these resolutions would have supported a further positive score update had the discussion not been frozen.**
>
>
> ---
>
> ### 4. Effect of the discussion freeze
>
> Because the process was frozen after our final clarifications and revision, the **current scores under-represent the final state of the discussion**:
>
> - **LWCT** actively engaged, called the rebuttal “pretty convincing overall,” asked for results on simpler backbones, and **raised their score from 4 to 6** after seeing the new GCN/GIN experiments on all datasets.
> - **h5ap** gave a **6** from the start and, after our detailed responses on discrete-time stability, the role of the energy, and the relation to Zhao et al. (NeurIPS 2023), raised no further concerns; there are **no unresolved issues** from h5ap at the time of the freeze.
> - **YfoY** rated soundness and presentation as **excellent**, found the decomposition and stability results appealing, and we answered all of their concrete questions (Euler stability, oversquashing wording, energy vs task loss, vanishing gradients). They did not get a chance to react to these clarifications or the updated text, and given the trajectory of the discussion, we believe their borderline-positive **4** would likely have been revised upward.
> - **sudZ** moved from **2 to 4** after our rebuttal and follow-ups on novelty, Lyapunov stability in GNNs, oversquashing, dataset choices, and why TANGO is graph-specific. The revised manuscript now includes exactly the comparisons and clarifications they requested, and we see no remaining unresolved concerns on their side.
>
> In short, for each reviewer we now have **either an explicit score increase or a complete written resolution of their concerns**; the discussion freeze prevented especially YfoY, and potentially sudZ, from updating their scores to reflect the fully resolved state of the paper.

---

> > ### Author Response · Authors · 2025-12-02
> > **Concluding Remarks by Authors (Part 3)**
> >
> > ### 5. Conclusion
> >
> > In summary, after rebuttal and discussion we have **two clear positive ratings of 6 (LWCT, h5ap)** and **two ratings of 4 (YfoY, sudZ)**, where **both 4s were increased from more negative starting points** during the discussion. For every reviewer, **all concrete concerns have been addressed directly and fully**, and the resulting clarifications, wording changes, and new experiments are incorporated in the current manuscript, including: discrete-time stability with Euler, the role of the learned energy, oversquashing claims and the interpretation of Proposition 4, positioning relative to Lyapunov-stable neural ODEs and energy-based GNNs, complexity/runtime/memory and hyperparameter sensitivity, backbone generality (GCN/GIN as well as GatedGCN/GPS) with targeted ablations, and the interaction between Lyapunov stability and long-range propagation.
> >
> > Reviewers who could not respond after our final updates (YfoY, h5ap, and sudZ for their final comments which were important according to them) leave **no unresolved technical objections** in the thread, and the direction of the discussion together with the upward score movements strongly suggests that their scores would likely have been further improved had the discussion not been frozen. The final paper thus presents a **principled Lyapunov-structured graph neural dynamics framework with a novel energy-plus-tangential decomposition**, carefully positioned within both the GNN and Lyapunov-stable NN literatures, and it demonstrates **strong empirical performance precisely on benchmarks where long-range propagation and oversquashing are most challenging**.
> >
> > We respectfully ask that you base your decision on this complete discussion record and the revised submission, and that you give *TANGO* serious consideration for acceptance.
> >
> > We thank you very much for your time and effort in handling our paper under the unusual ICLR 2026 circumstances.
> >
> > Sincerely,  and kindest regards,
> >
> > Authors

---

### Note · Program_Chairs · 2026-01-17
**Submission Desk Rejected by Program Chairs**

The following references in this submission do not refer to real documents and/or have major errors in bibliographic information:

 Sven Kreuzer, Michael Reiner, and Stefan D. D. De Villiers. Sant: Structural attention networks for graphs. Proceedings of the 38th International Conference on Machine Learning (ICML), 2021c.